# Contextual Policies Enable Efficient and Interpretable Inverse Reinforcement Learning for Populations

**Ville Tanskanen**                                          *ville.tanskanen@helsinki.fi*
*Department of Computer Science, University of Helsinki, Finland*

**Chang Rajani**                                             *chang.rajani@helsinki.fi*
*Department of Computer Science, University of Helsinki, Finland*

**Perttu Hämäläinen**                                        *perttu.hamalainen@aalto.fi*
*Aalto University, Finland*

**Christian Guckelsberger**                                  *christian.guckelsberger@aalto.fi*
*Department of Computer Science, Aalto University, Finland*
*School of Electrical Engineering and Computer Science, Queen Mary University of London, UK*

**Arto Klami**                                               *arto.klami@helsinki.fi*
*Department of Computer Science, University of Helsinki, Finland*

**Reviewed on OpenReview:** *https://openreview.net/forum?id=4CUkCG6ITe*

## Abstract

Inverse reinforcement learning (IRL) methods learn a reward function from expert demonstrations such as human behavior, offering a practical solution for crafting reward functions for complex environments. However, IRL is computationally expensive when applied to large populations of demonstrators, as existing IRL algorithms require solving a separate reinforcement learning (RL) problem for each individual. We propose a new IRL approach that relies on contextual RL, where an optimal policy is learned for multiple contexts. We first learn a contextual policy that provides the RL solution directly for a parametric family of reward functions, and then re-use it for IRL on each individual within the population. We motivate our method within the scenario of AI-driven playtesting of videogames, and focus on an interpretable family of reward functions. We evaluate the method on a navigation task and the battle arena game Derk, where it successfully recovers distinct player reward preferences from a simulated population and provides substantial time savings compared to a solid baseline of adversarial IRL.

## 1 Introduction

Modelling human behavior is one of the grand goals of AI, with versatile applications from Computational Cognitive Science to Human-Computer Interaction (Oulasvirta et al., 2022). User modelling in video games constitutes an example application in the latter domain and serves as the use-case for our contribution. Models of human behavior can implicitly inform game design choices, e.g. to design more engaging and satisfying games. Moreover, they can be explicitly employed for simulating player behavior, e.g. with the goal of reducing the need for costly, slow and tedious human playtesting sessions (Gudmundsson et al., 2018; Chang et al., 2019; Stahlke et al., 2020; Roohi et al., 2018).

Reinforcement learning (RL) provides one basis for building behavioral models. RL algorithms (Sutton & Barto, 2018) leverage an agent's experience in interacting with an environment to learn an optimal behavioral policy which maximizes a sum of future rewards. Whether or not humans truly behave as reward-maximizing agents, RL has been shown to be a useful framework (Silver et al., 2021), with empirical evidence

1) Contextual Reinforcement Learning     2) Fast Inverse Reinforcement Learning

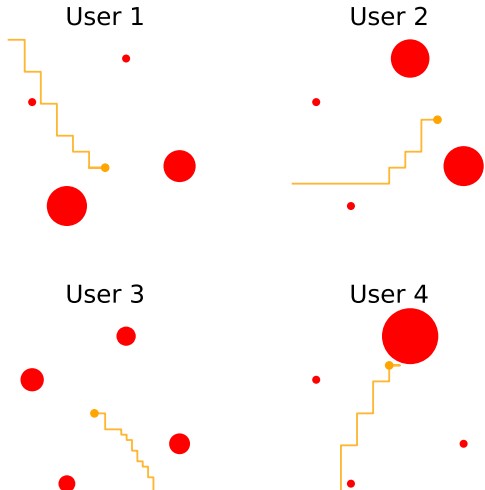

Figure 1: Efficient population-level IRL in a 2-D navigation task (see Section 4 for details). **Left**: We first find a contextual policy for a family of reward functions. These weight the preference of four target areas (red dots) without using the expert trajectories. The colored lines indicate the optimal trajectories for a range of preferences, all starting from the center for clarity. **Right**: For a set of expert trajectories (here one per user), we infer the user's reward preference (size of the target area) using a gradient-based algorithm that does not require solving any RL problems. Instead, the algorithm only needs the pre-computed contextual RL solution. This enables large-scale learning of users' preferences.

in behavioral modeling in many domains, including AI-driven playtesting (Bergdahl et al., 2020; Shin et al., 2020; Kristensen et al., 2020; Roohi et al., 2021; de Woillemont et al., 2021).

The key challenge in developing behavioral models with RL is in defining the reward function that matches an individual's goals. In some cases such reward functions can be derived directly from cognitive theories (Holmgård et al., 2014a;b), but a more general approach is to learn the reward function from observations of how the target individual interacts with the environment. The task of inferring the reward function from such *expert demonstrations* is called inverse reinforcement learning (IRL) (Abbeel & Ng, 2004), a sub-field of imitation learning (IL) (Osa et al., 2018; Pelling & Gardner, 2019). This formulation allows learning models even for an individual if given sufficient samples of their past behavior. Games serve as an excellent domain for studying and applying IRL, as they constitute worlds of arbitrary complexity and game companies often extensively log the gameplay of potentially millions of players.

In the current IRL literature, the rewards are often represented as flexible neural networks (Finn et al., 2016b; Fu et al., 2018) and the learning is often based on adversarial algorithms that contrast simulated and expert trajectories (Ho & Ermon, 2016; Fu et al., 2018; Wang et al., 2021). Despite notable progress in the core algorithms, all current solutions still scale poorly for large populations of demonstrators. In particular, vast majority of the methods require repeatedly solving the optimal policy within the IRL algorithms. When applied to large populations, repeatedly solving RL problems within iterative algorithms for each user is clearly wasteful. Earlier works that propose IRL methods that avoid using nested RL solutions still retain high cost for populations since they require training user-specific models instead of using a shared model (Klein et al., 2013; Sharma et al., 2017; Garg et al., 2021; Pirotta & Restelli, 2016).

Closest to our work, Ramponi et al. (2020) extended the gradient-based method of Pirotta & Restelli (2016) for populations by assuming that the demonstrators can be grouped to a few (in their experiments 2-3) clusters where all demonstrators within a cluster share the exact same reward. This improves efficiency when the clustering assumption holds, but does not help in learning individual rewards in general.

We contribute a method to speed up IRL for large populations based on the concept of *contextual RL* (cRL) (Hallak et al., 2015). Standard RL algorithms learn a policy for a specific environment with fixed (but often stochastic) transition dynamics and reward functions, whereas cRL methods learn policies for contextual environments where the dynamics and rewards are context-dependent. For instance, Eimer et al. (2021) learnt a contextual policy for a PointMass environment with parameterized friction, enabling the agent to behave optimally with arbitrary future friction coefficients.

Our core idea is to interpret a parameterization of the reward function as a context. This allows first learning a cRL solution for all possible reward preferences in advance, without yet accessing any expert demonstrations. We can then solve the IRL problem for individual users by leveraging the cRL solution. We thus separate the RL and IRL problem, overcoming the need to repeatedly execute an RL algorithm within the inner loop of IRL. This reduces the cost of the per-user computation at the expense of a fixed precomputation of the cRL policy. To the best of our knowledge, no prior work has used cRL for efficient population-level IRL, even though cRLs have recently been considered in related tasks. De Woillemont et al. (2021) employed cRL – without explicit reference to the concept – for AI playtesting, creating alternative player personas manually with the contextual policy. Merel et al. (2017) and Wang et al. (2017), in turn, used contexts for improving imitation of the demonstrations, without paying attention to the reward functions underlying the behavior. Finally, Moon et al. (2022, 2023) used cRLs as part of a behavioral model, but they do not solve the problem with IRL but rather rely on general-purpose inference methods. Contextual RL with reward as the context has been called *multi-objective RL* (MORL) in some works (Castelletti et al., 2012; Parisi et al., 2016; Yang et al., 2019). Many of these works focus on (using the terminology of cRL) learning the contextual policy and its theoretical aspects without considering the IRL task (Castelletti et al., 2012; Parisi et al., 2016). Closest to our work in this literature is Yang et al. (2019) where they introduce an efficient algorithm for learning contextual policies, and show how to recover hidden preferences (similar to IRL) with scalar rewards coming from simulator, only using the simulator and not demonstrations (differing from IRL).

Our approach is applicable to all behavioral modelling tasks, assuming a sufficiently low-dimensional parameterization for the reward function. We construct the reward function by specifying a set of sub-reward functions that are hypothesised to motivate the user's behavior, and consider cases where the sub-rewards are defined in advance as interpretable basis of behavior. Game companies often analyze their players through pre-defined metrics that characterise e.g. how often they use specific features of the game to obtain insights about their behavior on a population level. Such analysis, however, is necessarily limited to aggregate summaries and does not generalize for complex environments with no clear interpretation between rewards and game features or when reaching a particular reward requires a long sequence of actions. RL resolves these challenges as a natural basis of modelling sequential and context-dependent decisions, and the pre-existing metrics can directly be used as sub-rewards, requiring no additional developer effort. The full reward function is parametrized as a weighted combination of these sub-rewards, which is naturally low-dimensional and interpretable. However, the new IRL algorithm itself is general and applicable also for other reward functions.

We evaluate the approach in two simulation studies. We first employ it in a custom-made navigation task (Figure 1) to investigate the method's reliability in inferring the weights of a growing number of sub-rewards. We also compare our method to adversarial inverse reinforcement learning (AIRL) (Fu et al., 2018), showing that we require only 1% of the environment interactions for each user compared to AIRL, and hence show that dramatic speedups are possible for large populations even after accounting for the pre-training cost of cRL. Finally, we apply the method for distinguishing between players using either aggressive or passive playing strategies in `Derk` (Norén, 2020), a video game designed for developing RL algorithms.

## 2  Background

### 2.1  Reinforcement Learning

Reinforcement learning algorithms are formulated using *Markov Decision processes* (MDPs) (Sutton & Barto, 2018). A MDP describes an environment and task using a tuple $(\mathcal{S}, \mathcal{A}, P(s'|s, a), r)$, where $\mathcal{S}$ is the state space, $\mathcal{A}$ is the action space, $P(s'|s, a)$ is a distribution called *transition dynamics* that describes the evolution

of the system from any state $s \in \mathcal{S}$, to a future state $s' \in \mathcal{S}$, with an action $a \in \mathcal{A}$. The reward function $r : \mathcal{S} \times \mathcal{A} \times \mathcal{S} \to \mathbb{R}$ describes the utility of a transition $(s, a, s')$.

The goal is to find a policy $\pi : \mathcal{S} \to \mathcal{A}$ that maximizes the expected sum of (discounted) future rewards, also called expected return $J$. For stochastic environments and policies parameterized by $\boldsymbol{\theta}$ this corresponds to maximizing

$$J(\boldsymbol{\theta}) = \mathbb{E}_{\substack{s_{t+1} \sim p(s_{t+1}|a_t, s_t) \\ a_t \sim \pi_{\boldsymbol{\theta}}(s_t)}} \left[ \sum_{t=0}^{T} \gamma^t r(s_t, a_t, s_{t+1}) \right]. \tag{1}$$

where $\gamma$ is the discount factor. We use finite $T$ and hence consider episodic RL tasks. An episode is a collection of state-action-reward tuples starting from an initial state $s_0$ and terminating either at a time limit or a terminal state $s_T$.

The RL problem can be solved with numerous algorithms. For instance, when the transition dynamics are unknown we often employ *model-free* algorithms (e.g. Schulman et al. (2017)) and if the dynamics are known then *model-based* algorithms can be used (e.g. Chua et al. (2018)). Our work is agnostic w.r.t. the specific RL algorithm since our IRL approach merely requires running a suitable RL solver as an intermediate step, and hence the RL algorithm can be decided based on the properties of the environment.

## 2.2 Contextual Reinforcement Learning

Contextual reinforcement learning is about learning policies in contextual MDPs (cMDPs) (Hallak et al., 2015). It is used commonly in environments where the agent naturally faces multiple contexts, for instance different weather conditions in autonomous driving, or in difficult environments that can be parametrized in a way that curriculum learning methods can be employed to reach better final performances. Learning cRL policies is harder, as the environment becomes more complex due to the additional dependency on a context, but is proven to be doable (Jiang et al., 2017; Modi et al., 2018; Belogolovsky et al., 2021).

Formally, a contextual MDP is a tuple $(\mathcal{C}, \mathcal{S}, \mathcal{A}, \mathcal{M}(c))$, where $\mathcal{S}$ and $\mathcal{A}$ are state and action spaces like before, but $\mathcal{C}$ is a space of *contexts* and $\mathcal{M}(c)$ is a function from context space to a MDP $(\mathcal{S}, \mathcal{A}, P_c(s'|s, a), r_c)$, where the transition dynamics $P_c(s'|s, a)$ and the reward function $r_c$ depend on the context $c$.

We use a special case of the general cMDP where only the reward function $r_c$ is affected by the context and the transition dynamics are not; this means the context only encodes information about the reward function (and hence a given user, e.g. a player) rather than being a property of the environment (e.g. the game). This differs from many previous uses of cRL where the environment itself is contextual.

## 2.3 Inverse Reinforcement Learning

While the goal of RL is to find an optimal policy given the reward function, *inverse reinforcement learning* (IRL) refers to learning the reward function from expert demonstrations. We assume that every expert has a single reward $r_{\boldsymbol{w}}$ and the observed demonstrations correspond to a policy maximizing it. The reward function $r_{\boldsymbol{w}}$ is assumed to be parametrized by a vector $\boldsymbol{w}$, and the goal is to learn $\boldsymbol{w}$ such that a policy maximizing the corresponding return would be similar to the policy that generated the demonstrations. We assume access to a set of $N$ trajectories $\{\tau_i\}_{i=1}^N = \{\{s_0^{(i)}, a_1^{(i)}, \dots, s_{T_i}^{(i)}\}\}_{i=1}^N$, where $\tau_i$ is the $i$th trajectory and $s_t^{(i)}$ and $a_t^{(i)}$ are the state and action of the *$i$th* trajectory at timestep $t$, each trajectory being possibly of different length $T_i$. The goal is to maximize, w.r.t. the weights $\boldsymbol{w}$, the likelihood of $N$ independent expert demonstrations

$$G := P(\tau_1, \dots, \tau_N | \boldsymbol{w}) = \prod_{i=1}^{N} \frac{\exp\left(r_{\boldsymbol{w}}(\tau_i)\right)}{Z}, \tag{2}$$

where $Z = \int \exp\left(\sum_{t=0}^{T_i} r_{\boldsymbol{w}}(s_t, a_t, s_{t+1})\right) d\tau$ is the normalizing constant integrating over every possible trajectory sequence $\{s_0, a_1, s_1, \dots, s_T\}$. We also write $r_{\boldsymbol{w}}(\tau) = \sum_{t=0}^{T} r_{\boldsymbol{w}}(s_t, a_t, s_{t+1})$ for compactness. The form of the likelihood comes from the Boltzmann rationality assumption w.r.t. the demonstrations (Ziebart et al., 2008; Finn et al., 2016b); we assume the experts choose actions so that the ones yielding high returns are favored over others by an exponential factor.

---

**Algorithm 1:** Standard IRL

**1** Initialize $\boldsymbol{w}$ randomly;
**2** **for** $i = 0$ **to** $I$ **do**        `// IRL loop`
**3**    **for** $j = 0$ **to** $R$ **do**      `// RL loop`
**4**      $\pi_\theta \leftarrow$ Improve $\pi_\theta$ w.r.t. $\boldsymbol{\theta}$ using equation 1 and $r_{\boldsymbol{w}}$
**5**    **end**
**6**    $\boldsymbol{w} \leftarrow \boldsymbol{w} - \eta \frac{\partial \mathcal{L}(\boldsymbol{w})}{\partial \boldsymbol{w}}$;
**7** **end**

---

**Algorithm 2:** cRL + IRL

**1** **for** $j = 0$ **to** $R_c$ **do**        `// pretrain cRL`
**2**    $\boldsymbol{w} \sim \text{Dir}(\boldsymbol{\alpha})$;
**3**    $\pi_\theta(\cdot, \boldsymbol{w}) \leftarrow$ Improve $\pi_\theta(\cdot, \boldsymbol{w})$ w.r.t. $\boldsymbol{\theta}$ using equation 1 and $r_{\boldsymbol{w}}$
**4** **end**
**5** Initialize $\boldsymbol{w}$ randomly;
**6** **for** $i = 0$ **to** $I$ **do**    `// IRL with pretrained cRL`
**7**    $\boldsymbol{w} \leftarrow \boldsymbol{w} - \eta \frac{\partial \mathcal{L}(\boldsymbol{w})}{\partial \boldsymbol{w}}$;
**8** **end**

---

Since the integral in $Z$ is intractable, the objective of equation 2 is typically replaced by a surrogate loss

$$\mathcal{L}(\boldsymbol{w}) = \mathbb{E}_{\tau \sim \text{sg}(\pi(\tau|\boldsymbol{w}))} \left[ r_{\boldsymbol{w}}(\tau_i) \right] - \mathbb{E}_{\tau \sim \pi^*(\tau)} \left[ r_{\boldsymbol{w}}(\tau_i) \right], \tag{3}$$

whose gradient is equivalent to that of $-\frac{1}{N} \log G$, and where where $\text{sg}(\cdot)$ is the stop-gradient function. Here $\pi^*$ denotes the expert policy and $\pi(\tau|\boldsymbol{w})$ denotes the policy that is optimal w.r.t. the current parameters $\boldsymbol{w}$. The procedure can be expressed as a standard IRL algorithm, presented in Algorithm 1, where an RL policy is optimized for each IRL step to estimate the first expectation in equation 3 (cf. Algorithm 1, lines 3-5). Both IRL and RL are solved with iterative algorithms with $I$ and $R$ denoting the respective number of iterations. The exact updates depend on the algorithm.

In Section 4 we will compare our method against one commonly used IRL method, which relies on adversarial training. In adversarial IRL, two models are trained: a policy $\pi_\theta$ like before and a *discriminator $D$*, whose purpose is to classify trajectories (or state-action-next-state triplets) into two classes: those generated by the expert and those generated by the current policy. In *adversarial inverse reinforcement learning* (AIRL) (Fu et al., 2018), the discriminator takes the specific form of

$$D(s, a, s') = \frac{\exp(f_{\psi,\phi}(s, a, s'))}{\exp(f_{\psi,\phi}(s, a, s')) + \pi_\theta(a|s)}, \tag{4}$$

where $f_{\psi,\phi}(s, a, s') = g_\psi(s, a) + \gamma h_\phi(s') - h_\phi(s)$. Despite the adversarial formulation, the algorithm is closely related to the objective equation 3 (for details, see (Finn et al., 2016a) and (Fu et al., 2018)) and follows approximately Algorithm 1. The differences are that policy is trained to maximize $R(s, a, s') = \log(1 - D(s, a, s')) - \log(D(s, a, s'))$, that is, to fool the discriminator, while learning reward function is interchanged with discriminator training.

## 3 Method

### 3.1 Problem Setup

Our task is to solve the IRL problem for a population of users $u \in \{1, \ldots, U\}$, based on expert trajectories $\{\tau_i^u\}_{i=1}^{N_u}$ of each user. That is, we need to solve the optimization problem of equation 2 $U$ times. In many scenarios $U$ can be very large and for each user we may have a fairly large number of expert trajectories $N_u$. Hence, we aim to solve this problem in a computationally efficient manner. Furthermore, we seek a solution where the reward functions are (at least to some extent) interpretable. In our video game use case, this would provide developers insights on the player population to aid both game design and analytics.

### 3.2 Approach

#### 3.2.1 Overview

We propose an abstract general pipeline, where we combine cRL and IRL to overcome the computational burden of IRL. We first train a single contextual policy that is (sufficiently) optimal for any reward function characterized by the weights $\boldsymbol{w}$. This allows the inner loop of Algorithm 1 to be moved to a pre-training

phase, and the IRL loss in equation 3 to be calculated using the pre-trained contextual policy. Note that user trajectories are not used in the cRL part, but only revealed to the IRL part of the approach. Combining IRL with cRL can in principle speed up all IRL methods that require optimal policies for multiple reward functions. We provide technical details for one instance of the principle to demonstrate how it works, but note that other choices for the cRL and IRL training could be made. The procedure is outlined in Algorithm 2 and described in detail below.

### 3.2.2 Interpretable Reward Parameterization

We focus on linearly weighted sub-reward functions $[r^{(1)}(\cdot), \ldots, r^{(K)}(\cdot)]$, so that the reward function becomes

$$r_{\boldsymbol{w}}(s, a, s') = \sum_{k=1}^{K} w_k r^{(k)}(s, a, s'). \tag{5}$$

The sub-reward functions are assumed to be predefined by domain experts, and the IRL part only needs to learn the weights $\boldsymbol{w}$ that offer interpretability by providing information about user preferences among the sub-reward functions. This parameterization is similar to the early IRL works (Abbeel & Ng, 2004; Ziebart et al., 2008), but we additionally assume the weights $\boldsymbol{w}$ to be probability vectors, i.e. $\sum_{i=1}^{K} w_i = 1$, so that the individual weights can be interpreted as proportional importance of the sub-rewards.

Even though we run all experiments with this reward family motivated by our video game use case where the sub-rewards are pre-defined by the company as aspects they want to use as basis for understanding the individual user behavior, we note that the method itself is compatible with more general reward functions as long as we can solve the corresponding cRL problem. We discuss the feasibility of this in more detail in Section 5.

### 3.2.3 Contextual Policy Learning

A contextual policy depends not only on the states but also on the reward weights $\boldsymbol{w}$ that define the contexts here, making the policy $\pi : \mathcal{S} \times \mathcal{W} \rightarrow \mathcal{A}$ a function of state *and* reward weights as explained in Section 2.2. Importantly, in our formulation the weights $\boldsymbol{w}$ only influence the (user-dependent) reward function and not the rest of the environment.

As a specific choice for our pipeline, we choose to solve the cRL problem with the algorithm by de Woillemont et al. (2021) that iteratively improves the policy $\pi_{\boldsymbol{\theta}}(s, \boldsymbol{w})$ for randomly selected contexts $\boldsymbol{w}$. For each episode we sample new weights $\boldsymbol{w}$ and then improve the policy $\pi_{\boldsymbol{\theta}}(s, \boldsymbol{w})$ for this set of weights (lines 2-5 in Alg. 2). Note that other sampling strategies could be employed for training cRL as well, for example ones similar to Eimer et al. (2021) that try to sample contexts that are sufficiently difficult for the current policy.

We use neural networks (of two hidden layers in our experiments) for defining the contextual policy $\pi_{\boldsymbol{\theta}}(s, \boldsymbol{w})$. For updating the policy, we can use any iterative RL algorithm and we use the Proximal Policy Optimization (PPO) (Schulman et al., 2017) that has been widely adopted by the RL community. Even though $\boldsymbol{w}$ is kept fixed during an episode, the policies for other weights still change due to updating the shared policy parameters $\boldsymbol{\theta}$. This process is continued until the policy $\pi_{\boldsymbol{\theta}}(s, \boldsymbol{w})$ converges for all $\boldsymbol{w}$. In practice, we stop when the episode returns converge.

The weights $\boldsymbol{w}$ need to be probability vectors and are sampled from a Dirichlet distribution $\boldsymbol{w} \sim \text{Dir}(\alpha)$. In principle, any positive $\alpha$ would be applicable, but we found that $\alpha \rightarrow 0$ works well in practice [1] and provides an easy choice with no tunable hyperparameters. That is, we only select a single sub-reward at a time, but the cRL algorithm still learns to interpolate between these archetypal cases due to the continuous nature of neural networks that we use as policies.

Figure 1 (left) shows an example of the interpolation capabilities in a 2-D navigation task. During training, only trajectories aiming for a single goal were rewarded, but the solution still efficiently interpolates between the different goals. Note that this choice is made only for training the cRL, and the approach makes

---

[1]In early experimentation we found that allowing non one-hot vectors ($\alpha > 0$) led to the policy getting stuck in a local optimum where it only prefers one or few goals and others are ignored.

no assumptions on the distribution of $\boldsymbol{w}$ for the demonstrations. In all of the experiments the observed demonstrations correspond to $\boldsymbol{w}$ with multiple non-zero weights.

### 3.2.4 Gradient-based IRL

The converged contextual policy $\pi_\theta$ can be used to provide optimal behavior w.r.t. any weights $\boldsymbol{w}$ (that define reward function $r_{\boldsymbol{w}}$) by using the policy with fixed weights. This allows directly computing the IRL loss given by equation 3 in Line 7 of Algorithm 2, without requiring optimization of the policy.

The log-likelihood of equation 2 is maximized with gradient updates, using the standard gradients that now take the form

$$\nabla_{\boldsymbol{w}} \frac{1}{N} \log G = \mathbb{E}_{\tau \sim \pi^*(\tau)} \left[ \nabla_{\boldsymbol{w}} r_{\boldsymbol{w}}(\tau) \right] - \mathbb{E}_{\tau \sim \pi(\tau|\boldsymbol{w})} \left[ \nabla_{\boldsymbol{w}} r_{\boldsymbol{w}}(\tau) \right]. \tag{6}$$

The derivation is provided in the Appendix A. In practice we impose the simplex-constraint of $\boldsymbol{w}$ by optimizing the unconstrained weights $\boldsymbol{\nu} \in \mathbb{R}^K$ and transforming them by a softmax-operator $w_i = \frac{\exp(\nu_i)}{\sum_{j=1}^K \exp(\nu_j)}$

The softmax reparameterization introduces a non-identifiability (adding a constant value $c$ for each entry of $\boldsymbol{\nu}$ does not influence $\boldsymbol{w}$) which may in some cases introduce difficulties in optimization, but this formulation work well in our empirical experiments. Alternative techniques for accounting the constraint could be used as well.

The IRL problem is solved separately for each user, as in the standard approach, but the critical difference is that we now only need standard gradient descent, completely skipping the inner loop of $R$ iterations of the RL solver in Algorithm 1. The computational complexity for $U$ users and $I$ IRL iterations (per user) hence changes from $\mathcal{O}(UI(L+R))$ to $\mathcal{O}(R_c + UIL)$, where $L$ refers to the iterative computation that is required for the IRL loss, with substantial practical speedup for large $U$.

## 4 Experiments

We run experiments in two environments, an $m$-D navigation task and the video game Derk (Norén, 2020). The first affords straight-forward visualization and technical verification of the proposed approach, whereas the latter can demonstrate its applicability to more complex environments, in particular video games. We use simulated expert trajectories so that we have a ground truth for verifying the results, but the method is directly applicable to real user trajectories as well.

### 4.1 Experiment 1: m-D Navigation

#### 4.1.1 Purpose

We demonstrate that (a) the proposed approach is able to recover the true weights $\boldsymbol{w}$ from expert trajectories, and (b) that this works for a sufficiently large $K$, the number of sub-reward functions. We also show that our method learns the models for each individual in a fraction of the environment interactions compared to AIRL (Fu et al., 2018), a method shown to be highly competitive in recent IRL comparisons (Wang et al., 2021).

We use an environment that allows for easily modifying the number of sub-reward functions and measuring the quality of the recovered weights. Since we are assuming $K$ manually defined sub-rewards as a basis for interpretable reward functions, our aim is in the ballpark of approximately $K = 5$ and cases with $K \gg 10$ are not considered relevant.

#### 4.1.2 Environment and Task

We use an $m$-dimensional navigation task, where an agent must navigate towards a goal with discrete steps, similar to the environment used by Eimer et al. (2021). We extend this traditional environment by allowing multiple goals, each offering a distinct, proximity-based sub-reward function. We assume that the users may have different personal preferences for these sub-rewards, and hence the preferences define an optimal goal state in the environment. Figure 1 already illustrated a 2-D version of this environment and showed how

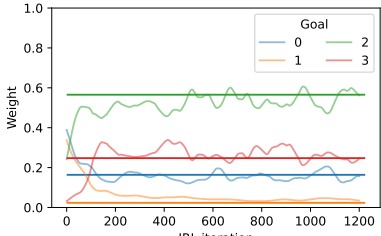 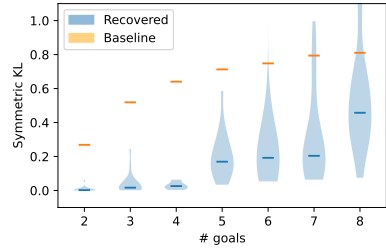 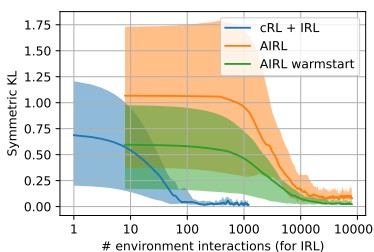

Figure 2: **Left:** An example convergence plot for real weights $[0.16, 0.02, 0.57, 0.25]$ (horizonal lines) and the corresponding recovered weights. The plot shows that, with a modest number of sub-rewards, our method recovers a good approximation of the behavior in a relatively small number of iterations. **Middle:** Symmetric KL divergences between the real weights and the recovered ones, averaged over 20 randomly sampled real weight vectors $\boldsymbol{w}$ for varying numbers of sub-rewards. We can correctly identify the weights up to $K = 4$ sub-rewards, after which the performance slowly deteriorates (and becomes theoretically unidentifiable at $K = 7$). We also show a naíve baseline prediction of $\frac{1}{K}$ proving a natural upper bound for the symmetric KL. **Right:** Sample complexity comparison of the proposed cRL + IRL method, AIRL and AIRL with warmstart for the IRL part of learning. The figure shows 0.25, 0.5 and 0.75-quantiles of symmetric KL over the 20 weight vectors related to 3 goals. Using cRL + IRL lowers the sample complexity compared to AIRL by two orders of magnitude. It also outperforms AIRL with warmstart by a similar magnitude.

the preferences influence the optimal policies. We note that, even though the environment may seem easy, it is relatively challenging from the RL perspective due to the high-dimensional action space and sparse state representation. We conduct experiments using $m = 5$ dimensions, which translates to $2m \times 4 \times 2 = 80$ possible discrete actions to choose from. Hence, solving the RL problem alone is non-trivial.

The environment can be formally described as (see Appendix B.4 for details):

- **States:** $s_t = [\boldsymbol{x}^{(a)}, \boldsymbol{x}^{(g_1)}, \ldots, \boldsymbol{x}^{(g_K)}, \boldsymbol{w}]$, where $\boldsymbol{x}^{(a)}$ and $\boldsymbol{x}^{(g_k)}$ are the $m$-dimensional vectors denoting the locations of the agent and the $k$th goal respectively, and $\boldsymbol{w}$ are the reward weights.

- **Actions:** $a = [a_1, a_2, a_3]$, $a_1$ for direction (discrete, $2m$ outcomes), $a_2$ for the stride (discrete, 4 outcomes) and $a_3$ does not affect the reward or the next state (nuisance).

- **Sub-rewards:** $r^{(k)}(s_t, a_t, s_{t+1}) = -||\boldsymbol{x}^{(a)}_{t+1} - \boldsymbol{x}^{(g_k)}||^2_2$

- **Goal locations:** The set of goals $\{g_k\}^K_{k=1}$ were chosen so that they are maximally far apart from each other on the $m$-dimensional 0.5-radius sphere. This ensures distinct goals and identifiability of rewards, so that for an environment of $m$ dimensions we can identify the behavior uniquely for a maximum of $m + 1$ goals.

- **Termination:** 150 steps, enough for reaching any goal.

We trained a cRL model for each choice of $K$, the number of goals. We then sampled 20 weight vectors $\{\boldsymbol{w}^{(\text{gen})}_i\}^{20}_{i=1}$ (denoting a user) for each $K \in \{2, \ldots, 8\}$, and generated 128 trajectories with each of them. The weight vectors $\boldsymbol{w}^{(\text{gen})}_i$ used for generating the trajectories were sampled from $\text{Dir}(\mathbf{1.0})$. These trajectories were then used as demonstrations for the IRL part to investigate how IRL performance is affected by increased number of sub-reward functions.

For the AIRL comparison, we use $K = 3$ goals with the same 20 weight vectors and trajectories, and compare the sample complexity required for a similar accuracy in weight recovery. To acquire linearly weighted rewards with AIRL, we define $g_\psi(s)$ to be the linear combination of weights and sub-rewards, and keep $h_\phi$ as a flexible neural network (like Fu et al. (2018) D.1).

We also compare to an intuitive extension of AIRL that attempts to leverage already trained models to speed up the learning for a new user. We are not aware of specific methods proposed for this, but introduce

here a simple variant that initialises the policy and discriminator of AIRL to a sensible checkpoint coming from the earlier AIRL run (specifically a checkpoint saved at 2000 environment iterations). This can be though of as observing part of the player population beforehand and using the solution learned for them as an initialization for future user-specific AIRL models. We call this method *AIRL warmstart*. Note, however, that the specific method is not to be interpreted as an optimal way of re-using previous computation, but is merely intended as an additional baseline.

### 4.1.3   Results

We inspect the results from three perspectives. Figure 2 (left) shows an example convergence of the weights over the IRL iterations, showing how the algorithm converges already within approximately 100-200 iterations and correctly identifies the preferences for a case of four sub-rewards. The weights were smoothed in a 50-step sliding window for visual clarity. The convergence for other ground truths and numbers of sub-reward functions in $K \in \{2, \ldots, 7\}$ is similar (see Appendix B for more examples).

Figure 2 (middle) quantifies the accuracy for varying numbers of sub-reward functions. We measure the goodness of recovered weights by calculating *symmetric Kullback-Leibler* (sKL) divergence (a.k.a the Jeffreys divergence, Jeffreys (1998); Kullback & Leibler (1951)) $D_{\mathrm{sKL}}(\boldsymbol{w}^{(\mathrm{gen})}, \hat{\boldsymbol{w}})$ between a generating weight vector $\boldsymbol{w}^{(\mathrm{gen})}$ and the recovered weight vector $\hat{\boldsymbol{w}}$. The sKL between two probability vectors $\boldsymbol{p}$ and $\boldsymbol{q}$ of length $K$ is defined by $D_{\mathrm{sKL}} := \sum_{i=1}^{K} p_i \log \frac{p_i}{q_i} + \sum_{i=1}^{K} q_i \log \frac{q_i}{p_i}$. For comparison, we also plot the expected sKL $\mathbb{E}_{\boldsymbol{p} \sim \mathrm{Dir}(\boldsymbol{\alpha})} \left[ D_{\mathrm{sKL}}(\boldsymbol{p}, \frac{1}{|\boldsymbol{p}|}) \right]$ denoting the symmetric KL for baseline predictions for weights, a natural upper bound for the divergence. Symmetric KL measure was chosen as our weights carry a probability interpretation, and the KL divergence is naturally comparable between varying length vectors. Other common metrics could have been used, and the results were similar for example for Euclidean distance between log-weights.

The results show that we can accurately estimate the true weight vectors up to $K = 4$. For $K > 4$ the performance starts to deteriorate because the problem becomes harder but we still clearly improve over the baseline of mean prediction. For $K \geq 7$ the problem is no longer identifiable for the $m = 5$ dimensional environment since multiple weight vectors can explain the same trajectories equally well (see Appendix C), and the performance indeed drops. In other words, we verify that we can solve the IRL problem when it is solvable, but also highlight that the designer needs to be careful in determining the sub-rewards in a way that is appropriate for the scenario. For scenarios with non-identifiable sub-rewards (e.g. the same action always provides the same two sub-rewards) it is impossible to recover unique weights.

However, even in those cases we can verify that the policies $\pi(\cdot, \hat{\boldsymbol{w}})$ induced by the recovered weights are meaningful, by comparing the actual returns. The results presented in Appendix D confirm that the policies using the recovered weights achieve similar performance to the policies using the ground truth weights $\boldsymbol{w}^{(\mathrm{gen})}$, even for large $K$'s.

Finally, Fig. 2 (right) compares the convergence of the proposed method against AIRL (Fu et al., 2018), a representative example of how IRL problems are solved in the literature. Averaged over the 20 ground truth weight vectors, our method converges in approximately 100 environment interactions for each user, whereas AIRL requires approximately 100 times more interactions due to repeatedly solving an RL problem within the algorithm (cf. Section 2). That is, we can solve approximately hundred IRL problems at the same time it takes to run AIRL once. The AIRL warmstart method that attempts to leverage previous computation in an alternative way leads to better initial performance, but takes just as long as standard AIRL to converge. For fair comparison we need to account for the pre-training cost needed for cRL. We used 2 million interactions for that, but in practice the method converged after 500K interactions (see Appendix B.4). This is approximately the same cost as solving 50 IRL problems with AIRL. For $U < 50$ AIRL would hence be faster, but for $U \gg 50$ we get a substantial speedup.

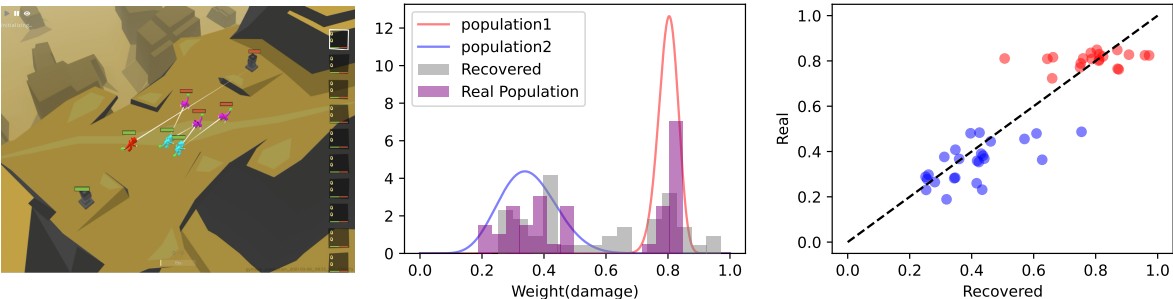

Figure 3: **Left:** Screenshot of the Derk environment (Norén, 2020) used for the experiments. The red agent is controlled, cyan is its team, and purple is the opposing team. **Middle:** Weights for the damage sub-reward $r_{\mathrm{dmg}}$ for a population of 45 simulated players in the Derk game. The two populations of players are easily recognizable from the recovered weights. **Right:** Cross-plot between recovered and real weights for the damage reward $r_{\mathrm{dmg}}$ (correlation coefficient 0.9).

## 4.2 Experiment 2: Population IRL for Derk

### 4.2.1 Purpose

To demonstrate the feasibility of the method for video game player modelling with large populations, we use the Multiplayer online battle arena (MOBA) style game `Derk` (Norén, 2020), developed for testing RL algorithms in a realistic environment, and apply the proposed method to discover clusters of player behaviors similar to Babes et al. (2011). We design two sub-reward functions, corresponding to dealing damage and healing, and show that we can reliably infer the weights of (simulated) users based on a reasonable amount of expert trajectories for each user. Note that the clustering assumption is here made only to simplify the data generation process and analysis; the model still learns individual rewards for each user. For this specific setup also methods that explicitly cluster the population could be used (Ramponi et al., 2020), but they explicitly leverage the clustering assumption that our method does not make.

### 4.2.2 Environment and Task

`Derk` (Norén, 2020) involves two teams, each having one tower to protect and three agents capable of moving, healing and attacking other agents – see Figure 3 (left) for a screenshot of the game. We provide here a high-level description of the environment as used in our experiments, with the details Appendix B.5.

We learn the policy for a single agent, treating the other agents as part of the environment. A 64-dimensional state vector describes the state of the environment, including e.g. the agents' positions, orientations and whether an attack is available. We set a time limit of 150 steps and assume that all agents have the same abilities: a melee attack that requires being near the enemy, and a healing ability that can be used for healing from a distance. Finally, we discretized the action space to be a discrete-valued vector with 5 dimensions for movement (turning and advancing), focus, attack and heal.

We use $K = 2$ intuitive sub-reward functions: $r_{\mathrm{dmg}}$ rewarding for dealing damage on the enemies and $r_{\mathrm{heal}}$ for healing a teammate. Learning the weights on these sub-reward functions would allow a game designer to e.g. identify whether there are players who tend to ignore safety of the teammates and are focusing on personally killing the opponents. Notably, the healing action is only rewarded in case the player is sufficiently close to an injured teammate, and hence even users with a high preference for the healing action need to get involved in combat. This makes identifying the preference for healing difficult.

The bots that were not the target of the optimization (Figure 3 left, cyan and purple) were controlled by a separate *bot policy*. During the cRL training, this was the same policy as the one being trained, copied to the bots after every 20 update steps. This is a form of self-play, that has been successfully used for example by Silver et al. (2017). The bot policy also sampled a new set of weights every episode.

During the IRL training we used a different bot policy to ensure that the approach is not overfitting to the policy used for the cRL: a simple heuristic where the bots walk toward the nearest enemy at every timestep and melee attack them if it is closer than a fixed threshold. The cRL converged in 11 hours and each user's IRL computation took approximately 11 minutes on single machine with P100 GPU.

### 4.2.3 Expert Trajectories

We generate $U = 45$ distinct weight vectors $\boldsymbol{w}$ and simulate a different playing style with each one of them. To do so, we first learn a cRL solution that provides a policy $\pi(s, \boldsymbol{w})$ for all $\boldsymbol{w}$ and then use the policy to simulate the trajectories.

More specifically, we sample $N_1 = 20$ weights from $\boldsymbol{w}^{(\mathrm{pop1})} \sim \mathrm{Beta}(127.2,\ 31.8)$ and $N_2 = 25$ weights from $\boldsymbol{w}^{(\mathrm{pop2})} \sim \mathrm{Beta}(9.6,\ 17.8)$ to represent two sub-populations of users (Figure 3, middle). These distributions were chosen by fixing the means to be distinct enough and variances so that the distributions have some overlap, and then solving for the parameters. For each simulated user we generate 128 trajectories (rounds), used for solving the IRL problem.

### 4.2.4 Results

We evaluate the method by comparing the estimated weights for the damage reward to the true ones used for generating the expert trajectories. The Pearson correlation between these two is $\rho = 0.9$ and the mean absolute error is 0.08, confirming that we can estimate the true weights with sufficient accuracy. The results are illustrated in Figure 3. The middle sub-plot shows histograms of the generated and recovered weights, confirming that we can recover the sub-populations that are interpreted as players who focus on killing the opponents (population 1) and players that focus on keeping their own team alive (population 2). The right sub-plot shows a cross-plot of the *individual* weights.

## 5 Discussion

Our main contribution is the new approach and hence the experiments focus on validating its behavior. The `Derk` game – even in the simplified format considered here – is a complex environment that resembles well the typical intended use case and is more complex than e.g. many single-player mobile games, but the main limitation of the experiments is the reliance on simulated users. We focused on introducing the cRL+IRL approach and demonstrating its feasibility; obtaining data from a commercial game for user experiments is subject to more application-oriented future work.

Despite the focus on large-scale population tasks, we only used 45 users in the `Derk` experiment and 20 users (for each $K$) in the navigation experiment to avoid an excess waste of energy – in both cases the cost scales linearly as a function of the users as the IRL problems are solved independently and running the experiments explicitly for larger populations would not provide any additional value. For the navigation environment, we showed that the proposed method is orders of magnitude better than the baseline method AIRL in terms of sample complexity after the cRL pre-training. We decided not to train the baseline method for `Derk` which is considerably slower due to rendering the screen after each step, but note that the per-user speedup would be dramatic. In this environment already a single standard RL solver takes approximately 2 hours and hence AIRL and other standard IRL methods that solve it repeatedly would take considerably longer, yet we get the solution for each user in 11 minutes. The exact speedup depends on the context, but should be of similar order for all environments and against all standard IRL algorithms that need to solve RL problems within IRL loop.

From the perspective of interpretability, it is worth noting that we cannot always learn a unique solution. For the navigation task we specifically constructed the environment and the sub-rewards so that the problem is identifiable (see Appendix B.4) but for general environments and sub-rewards this cannot be ensured, and even degenerate solutions are possible. For example, if one of the sub-rewards is constant we can optimize equation 3 by setting its weight to one (see Appendix A). Recently, Cao et al. (2021), Kim et al. (2021) and Metelli et al. (2023) studied the identifiability of IRL in general, and while their analysis is done for discrete state-action spaces, it is likely to generalize for our setup: Additional assumptions on the environment

(e.g. on sub-rewards, transition dynamics, expert trajectories and/or discount factors) are required to attain proven identifiability. Our reward function family was motivated by personal discussion with game developers who already have concrete sub-rewards (e.g. different types of resources) for their games, and who want to investigate the players from the perspective of these sub-rewards. We hence assumed the rewards have been chosen with reasonable care, for instance avoiding degenerate scenarios, and that we have observed sufficiently many interactions to infer the reward preference. We leave recommendations on how the sub-rewards should be designed if interested in uniquely defined solution as future work.

When interpretability of the users's reward functions is not important and the goal is to simulate users' behavior, then a more flexible reward structure, such as neural network, may be preferable because of its stronger expressive power. Algorithm 2 generalizes directly for such reward functions and does not require any changes in implementation, but proper experimentation would naturally be needed to validate how it works in practice. Learning the contextual policy for non-linear rewards is more challenging, but Sodhani et al. (2021) have shown that the cRL problem can be solved even in cases where the context is a 768-dimensional encoding of natural language, and methods like curriculum learning (Klink et al., 2020; Eimer et al., 2021) could be useful for learning the cRL policy in these situations. In our work, the context corresponds to the parameterization of the reward function, and hence these results suggests that we could use (reasonably-sized) neural networks as a family for the reward functions and could still learn the cRL. However, the distances $d(\boldsymbol{w}, \hat{\boldsymbol{w}})$ between real and recovered parameters would no longer be informative as quality measures due to non-identifiability of neural networks, and hence the generated behaviors would need to be compared instead. Devising a suitable metric for these kinds of trajectory comparisons is a core challenge in many imitation learning problems (Ho & Ermon, 2016; Fu et al., 2018; Wang et al., 2021; Ciosek, 2022).

## 6 Conclusion

We presented a novel integration of contextual reinforcement learning and inverse reinforcement learning to overcome prohibitive computational complexity of IRL for large populations. This was achieved by replacing repeated and user-specific RL solvers within an IRL algorithm with general contextual RL solution for a family of possible reward functions, trained only once without needing access to the user demonstrations. We showed this lowers the total sample complexity needed for solving the IRL problem for large user populations by orders of magnitude. The approach is agnostic to the specific RL algorithms. In this work we introduced one practical instance and studied how the number of sub-reward functions affects its performance. We also empirically demonstrated that we can correctly identify simulated player profiles in a realistic MOBA game environment.

Our work opens new possibilities for inverse reinforcement learning at a scale applicable in industry. We here considered an application case in game industry where companies want to offer personalized experiences for their players and understanding their behavior as individuals is the first step for customized games. However, any domain where a designer might want to study and understand user behavior and/or optimize the experience of an individual user or a specific user group can benefit from the introduced method.

**Acknowledgments**

This work was supported by Business Finland (project MINERAL) and the Academy of Finland (Flagship programme: Finnish Center for Artificial Intelligence, FCAI). The authors wish to thank the Finnish Computing Competence Infrastructure (FCCI) for supporting this project with computational and data storage resources.

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

## Appendix

## A  Gradient of the IRL Likelihood

The likelihood of the expert demonstrations for the IRL was given in equation 3. The scaled gradient in equation 6 follows from

$$\nabla_{\boldsymbol{w}} \frac{1}{N} \log G = \nabla_{\boldsymbol{w}} \frac{1}{N} \left( \sum_{i=1}^{N} \sum_{t=0}^{T} r_{\boldsymbol{w}}(s_t^{(i)}, a_t^{(i)}) - N \log Z \right)$$

$$= \nabla_{\boldsymbol{w}} \left( \frac{1}{N} \sum_{i=1}^{N} r_{\boldsymbol{w}}(\tau_i) - \log Z \right) = \frac{1}{N} \sum_{i=1}^{N} \nabla_{\boldsymbol{w}} r_{\boldsymbol{w}}(\tau_i) - \nabla_{\boldsymbol{w}} \log Z$$

$$= \frac{1}{N} \sum_{i=1}^{N} \nabla_{\boldsymbol{w}} r_{\boldsymbol{w}}(\tau_i) - \frac{1}{Z} \nabla_{\boldsymbol{w}} \int \exp\left( r_{\boldsymbol{w}}(\tau) \right) d\tau$$

$$= \frac{1}{N} \sum_{i=1}^{N} \nabla_{\boldsymbol{w}} r_{\boldsymbol{w}}(\tau_i) - \frac{1}{Z} \int \exp\left( r_{\boldsymbol{w}}(\tau_i) \right) \nabla_{\boldsymbol{w}} r_{\boldsymbol{w}}(\tau_i) d\tau$$

$$\approx \mathbb{E}_{\tau \sim \pi^*(\tau)} \left[ \nabla_{\boldsymbol{w}} r_{\boldsymbol{w}}(\tau_i) \right] - \int p(\tau|\boldsymbol{w}) \nabla_{\boldsymbol{w}} r_{\boldsymbol{w}}(\tau_i) d\tau$$

$$= \mathbb{E}_{\tau \sim \pi^*(\tau)} \left[ \nabla_{\boldsymbol{w}} r_{\boldsymbol{w}}(\tau_i) \right] - \mathbb{E}_{\tau \sim \pi(\tau|\boldsymbol{w})} \left[ \nabla_{\boldsymbol{w}} r_{\boldsymbol{w}}(\tau_i) \right], \tag{7}$$

where individual steps correspond to standard algebraic manipulation, or replacing empirical averages with expectations. This gradient is used for updating the weights $\boldsymbol{w}$ of the sub-reward functions in gradient-based optimization algorithms by employing a surrogate loss function

$$\mathcal{L}(\boldsymbol{w}) = \mathbb{E}_{\tau \sim \text{sg}(\pi(\tau|\boldsymbol{w}))} \left[ r_{\boldsymbol{w}}(\tau_i) \right] - \mathbb{E}_{\tau \sim \pi^*(\tau)} \left[ r_{\boldsymbol{w}}(\tau_i) \right]. \tag{8}$$

If we assume that the experts behave according to our parameterized policy with weights $w^*$, we can replace $\pi^*(\tau) = \pi(\tau|w^*)$, and it is easy to see that value of the loss at this (desired) point is $\mathcal{L}(w^*) = 0$.

We remark that in specific pathological cases we can reach zero loss also with solutions that are not interesting from the perspective of IRL. In particular, if we have a sub-reward $r_k$ such that $r_k(s, a, s') = c$ is constant, and $w_k = 1.0$, then the loss becomes

$$
\begin{aligned}
\mathcal{L}(\boldsymbol{w}) &= \mathbb{E}_{\tau \sim \mathrm{sg}(\pi(\tau|\boldsymbol{w}))} \left[ \sum_{t=1}^{T_\tau} r_k(s_t, a_t, s_t + 1) \right] - \mathbb{E}_{\tau \sim \pi^*(\tau)} \left[ \sum_{t=1}^{T_\tau} r_k(s_t, a_t, s_t + 1) \right] \\
&= \mathbb{E}_{\tau \sim \mathrm{sg}(\pi(\tau|\boldsymbol{w}))} [T_\tau c] - \mathbb{E}_{\tau \sim \pi^*(\tau)} [T_\tau c] \\
&= c \left( \mathbb{E}_{\tau \sim \mathrm{sg}(\pi(\tau|\boldsymbol{w}))} [T_\tau] - \mathbb{E}_{\tau \sim \pi^*(\tau)} [T_\tau] \right) \\
&= 0,
\end{aligned}
$$

if we assume equal length trajectories. That is, we can optimize the objective by assigning all weight on that constant reward. This problem case is easily avoided by not using constant sub-rewards, but exact characterization of conditions for which the correct IRL solution is the only global optimum of the objective remains open.

## B   Experimental Details

### B.1   PPO hyperparameters

We used Proximal Policy Optimization (PPO) (Schulman et al., 2017) algorithm for updating the contextual policy. PPO works by running a policy in the environment, collecting $N$ episodes of states, actions, rewards and action probabilities from $N_{\mathrm{arenas}}$ parallel environments, and then updating the policy with the gathered episodes. Like all actor-critic algorithms, the method contains a policy $\pi$ (actor) that is used to compute actions and a value estimator $\hat{V}$ (critic) that is used for evaluating rewards-to-go for a state. The loss function that PPO minimizes is given by

$$
-\mathbb{E} \left[ \min \left( r_t(\theta) \hat{A}_t, \mathrm{clip}(r_t(\theta), 1 - \epsilon, 1 + \epsilon) \hat{A}_t \right) \right] + c_1 \mathbb{E} \left[ \left( \hat{V}(s_t) - V(s_t) \right)^2 \right] - c_2 \mathbb{E} \left[ \log \pi_\theta(a_t|s_t) \right], \quad (9)
$$

where $\hat{A}_t$ is the advantage estimator, $r_t(\theta) = \frac{\pi_\theta(a_t|s_t)}{\pi_{\theta_{\mathrm{old}}}(a_t|s_t)}$ is the ratio of action probabilities computed by comparing action probabilities given by a network with current parameters $\theta$ with those computed with $\theta_{\mathrm{old}}$, the parameters that were used for generating the episodes [2]. We compute the advantages using Generalized Advantage Estimator (GAE-)$\lambda$ method Schulman et al. (2015), where the advantage $\hat{A}_t = \delta_t + (\gamma\lambda)\delta_{t+1} + \ldots + (\gamma\lambda)^{T-t+1}\delta_{T-1}$ is calculated using a value function estimator $\hat{V}$, discount factor $\gamma$ and GAE parameter $\lambda$ and where $\delta_t = r_t + \gamma\hat{V}(s_{t+1}) - \hat{V}(s_t)$. Value function estimator $\hat{V}(s_t)$ is trained to estimate the rewards-to-go $V(s_t)$ by using a mean squared error loss, weighted by a constant $c_1$ (second term of the loss). The entropy term (third term) is maximized to encourage as much randomness as possible. The effect of entropy is weighted by a factor of $c_2$.

We evaluate the loss in equation 9 with mini-batches of size $N_{\mathrm{batch}}$. The parameters of the actor and critic networks are optimized with Adam optimizer (Kingma & Ba, 2014) using learning rates $\alpha_{\mathrm{actor}}$ and $\alpha_{\mathrm{critic}}$ respectively. The optimization is run for $N_{\mathrm{epoch}}$ epochs for every $N$ episodes collected. Optimization is halted when a total of $N_{\mathrm{total}}$ timesteps is reached. The hyperparameter values that were used to train the cRL policy in our experiments are given in Table 1.

### B.2   Inverse Reinforcement Learning

The IRL hyperparameters of our method are provided in Table 2. The IRL part of Algorithm 2 is run for $I$ iterations, optimizing equation 3 with Adam (Kingma & Ba, 2014) optimizer using $\alpha_{\mathrm{IRL}}$ as the learning rate. The expectations in equation 3 are evaluated using batches of sizes $N_{\boldsymbol{w}}$ and $N_{\mathrm{expert}}$ respectively. To evaluate the first term of equation 3 we use $N_{\boldsymbol{w}}$ of most recent simulated trajectories, while for the second expectation we use $N_{\mathrm{expert}}$ randomly sampled expert trajectories. We simulate $N_{\mathrm{sim}}$ trajectories per IRL iteration and update the weights $\boldsymbol{w}$ using a batch of expert and simulation trajectories for $U_{\mathrm{IRL}}$ iterations.

---

[2]Remember that we update the policy multiple times, so when we begin updating, the first iteration has $\theta = \theta_{\mathrm{old}}$

|  |  | 5-D Nav. | Derk |
|---|---|---|---|
| Total timesteps | $(N_{\text{total}})$ | 2e6 | 5e5 |
| Arenas | $(N_{\text{arenas}})$ | 1 | 32 |
| Episodes per update | $(N)$ | 3 | 4 |
| Epohcs | $(N_{\text{epoch}})$ | 35 | 20 |
| Batch size | $(N_{\text{batch}})$ | 2048 | 4096 |
| KL clip fraction | $(\epsilon)$ | 0.125 | 0.2 |
| Actor loss coefficient | $(c_1)$ | 0.5 | 0.5 |
| Entropy coefficient | $(c_2)$ | 0.031 | 0.01 |
| Actor learning rate | $(\alpha_{\text{actor}})$ | 0.005 | 0.003 |
| Critic learning rate | $(\alpha_{\text{critic}})$ | 0.021 | 0.005 |
| Discount factor | $(\gamma)$ | 0.91 | 0.99 |
| GAE-$\lambda$ | $(\lambda)$ | 0.959 | 0.95 |

Table 1: Hyperparameters used for training the cRL

|  |  | 5-D Nav. | Derk |
|---|---|---|---|
| Iterations | $(I)$ | 1200 | 40 |
| Learning rate | $(\alpha_{\text{IRL}})$ | 0.01 | 0.1 |
| Env. simulation | $(N_{\text{sim}})$ | 1 | 32 |
| Weight batch | $(N_{\boldsymbol{w}})$ | 8 (last) | 32 (last) |
| Expert batch | $(N_{\text{expert}})$ | 32 | 32 |
| Updates per iteration | $(U_{\text{IRL}})$ | 2 | 1 |

Table 2: Hyperparameters used for training the IRL

The baseline that we compare against, *adversarial inverse reinfrocement learning* (AIRL) (Fu et al., 2018) is an established adversarial method that learns by iterating between training a discriminator and training a policy. Table 3 provides all of the hyperparameters used for training AIRL. We use PPO as the policy training algorithm, for which the hyperparameter names and descriptions are the same as for learning the cRL policy (Table 1). We train the AIRL for $N_{\text{AIRL}}$ iterations, where we first simulate $N_{\text{sim}}$ environment trajectories with the current policy. We than sample a batch of $N_{\boldsymbol{w}}$ simulated trajectories, called the simulation batch, and $N_{\text{expert}}$ trajectories for expert batch. Then, the discriminator is updated using binary cross-entropy loss with simulation batch labelled with negative labels, and expert batch labelled with positive labels using AdamW optimizer (Loshchilov & Hutter, 2019) with smoothing parameters $\beta = [0.5, 0.999]$ and learning rate of $\alpha_D$. After the discriminator update, the policy is updated for $N_{\text{epochs}}$ using batch size of $N_{\text{batch}}$ using the same procedure as described in Section B.1.

For the $m$-D navigation experiment we found the recovered weights to oscillate around the correct values (see. Fig. 4) because of two things: a) We used off-policy samples to estimate the first term of equation 3 (see Table 2: $N_{\text{sim}} = 1$ but $N_{\boldsymbol{w}} = 8$, meaning one on-policy and 7 off-policy samples from the last simulations are used) and b) because of the theoretical minimum of the IRL loss in equation 3 being 0, but the loss allowing negative values as well. We clamped the IRL loss on $m$-D navigation environment to mitigate (although not completely removing) the oscillation so that the clamped loss becomes $\tilde{\mathcal{L}}(\boldsymbol{w}) = \max(0, \mathcal{L}(\boldsymbol{w}))$. Clamping would have been possible also on Derk environment, but due to the parallel computation of the environment we only used on-policy samples for evaluation and opted out from clamping the IRL loss.

### B.3 Hyperparameter Selection

The values for the hyperparameters used in cRL (Table 1) were initially randomly searched in 2-d navigation task to get an initial guess of good values. Then for each experiment, they were manually tuned during preliminary experimentation. The values for IRL experiment (Table 2) were manually tuned by inspecting at the convergences of the losses and the weights with a random seed. The values for AIRL (Table 3) were random searched using trajectories related to a single weight vector.

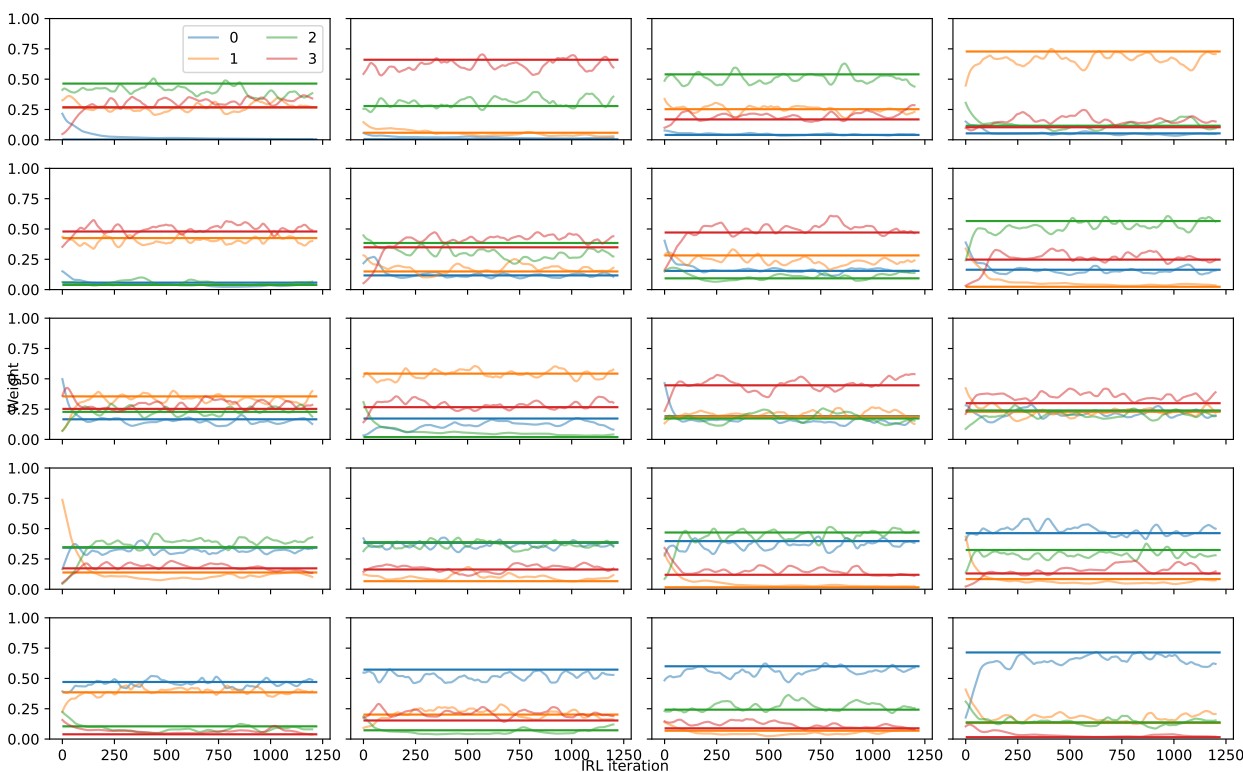

Figure 4: All 20 convergence plots for $K = 4$ goals for the IRL part.

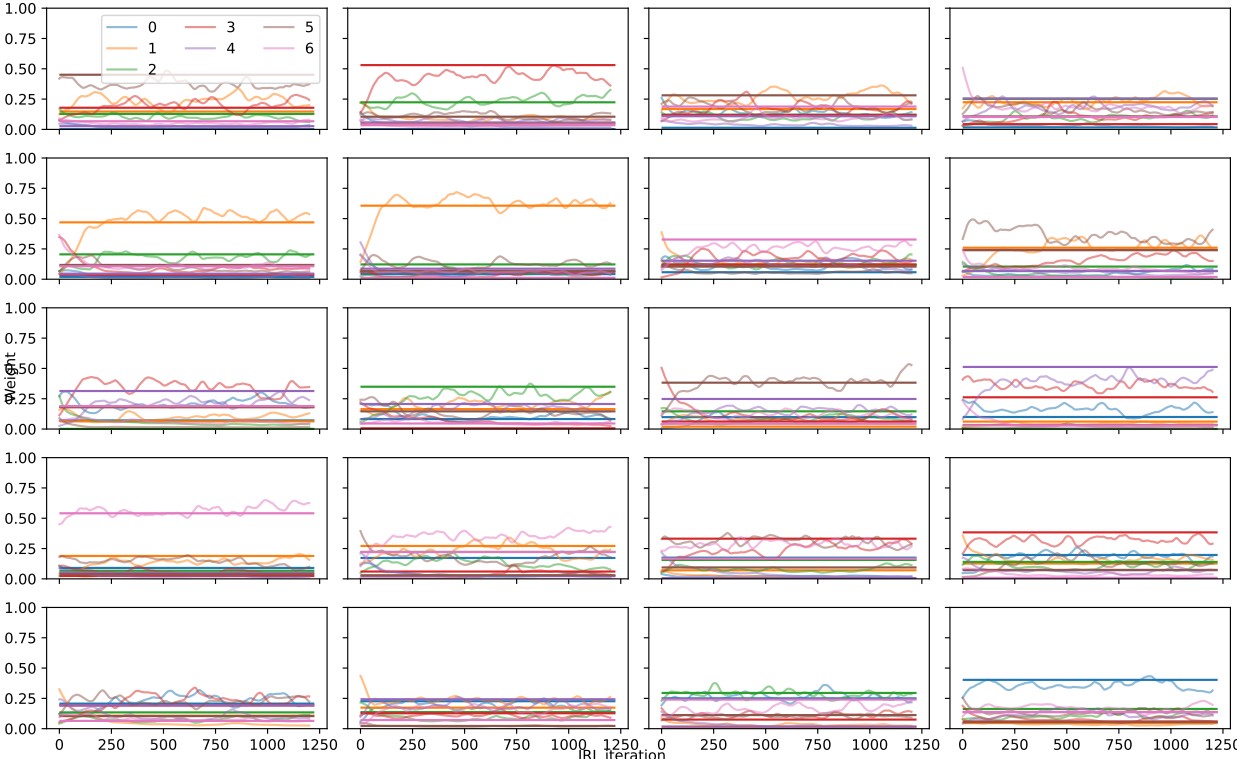

Figure 5: All 20 convergence plots for $K = 7$ goals for the IRL part.

|  |  | 5-D Nav. |
|---|---|---|
| Iterations | $(N_{\mathrm{AIRL}})$ | 10000 |
| Episodes per update | $(N)$ | 8 |
| Epohcs | $(N_{\mathrm{epoch}})$ | 3 |
| Batch size | $(N_{\mathrm{batch}})$ | 150 |
| KL clip fraction | $(\epsilon)$ | 0.2 |
| Actor loss coefficient | $(c_1)$ | 0.5 |
| Entropy coefficient | $(c_2)$ | 0.01 |
| Actor learning rate | $(\alpha_{\mathrm{actor}})$ | 0.0003 |
| Critic learning rate | $(\alpha_{\mathrm{critic}})$ | 0.001 |
| Disc. learning rate | $(\alpha_D)$ | 0.001 |
| Discount factor | $(\gamma)$ | 0.99 |
| GAE-$\lambda$ | $(\lambda)$ | 0.95 |
| Env. simulation | $(N_{\mathrm{sim}})$ | 8 |
| Weight batch | $(N_{\boldsymbol{w}})$ | 40 (last) |
| Expert batch | $(N_{\mathrm{expert}})$ | 32 |
| Disc. regularizer | $(c_3)$ | 0.01 |

Table 3: Hyperparameters used for the AIRL baseline. The PPO hyperparamters match the ones in Table 1.

### B.4   m-D Navigation

#### B.4.1   Goal locations

The set of goals $\{g_k\}_{k=1}^K$ in $m$-D navigation task were chosen so that they are maximally apart from each other on the $m$-dimensional 0.5-radius circle.

In particular, we sampled 1000 sets $K$ locations $\{\boldsymbol{x}_s^{(g_1)}, \ldots, \boldsymbol{x}_s^{(g_K)}\}_{s=1}^{1000}$ and chose the set that had the largest minimum pairwise distance. That is,

$$\max_s \min\{ \, ||\boldsymbol{x}_s^{(g_i)} - \boldsymbol{x}_s^{(g_j)}||_2 \mid i, j \in \{1, \ldots, K\}, i \neq j \, \}.$$

The reason for the sample-based approach is that placing equidistant points on a $m$-ball proves to be a non-trivial problem in spaces larger than 2-dimensional.

#### B.4.2   Training cRL

Contextual RL policy is trained in a pre-training phase, and for fair comparison of the sample complexities of different methods, the pretraining needs to be accounted for. We compare the complexity in $m = 5$ - dimensional navigation environment and show the training curves of cRL for $K \in \{2, \ldots, 8\}$ goals in Figure 6. We see that the pretraining converges around 500K time steps. which is equivalent of approx. 50 AIRL computations, making our proposed approach faster for populations larger than this. The plot highlights $K = 3$ goals, as the comparison was done on environment with three goals.

### B.5   Derk environment

#### B.5.1   State

The state of the original Derk contained the following 64 features: `Hitpoints, Ability0Ready, FriendStatueDistance, FriendStatueAngle, Friend1Distance, Friend1Angle, Friend2Distance, Friend2Angle, EnemyStatueDistance, EnemyStatueAngle, Enemy1Distance, Enemy1Angle, Enemy2Distance, Enemy2Angle, Enemy3Distance, Enemy3Angle, HasFocus, FocusRelativeRotation, FocusFacingUs, FocusFocusingBack, FocusHitpoints, Ability1Ready, Ability2Ready, FocusDazed, FocusCrippled, HeightFront1, HeightFront5, HeightBack2, PositionLeftRight, PositionUpDown, Stuck, UnusedSense31, HasTalons, HasBloodClaws, HasCleavers, HasCripplers, HasHealingGland, HasVampireGland, HasFrogLegs, HasPistol, HasMagnum, HasBlaster, HasParalyzingDart, HasIronBubblegum, HasHeliumBubblegum, HasShell, HasTrombone, FocusHasTalons,`

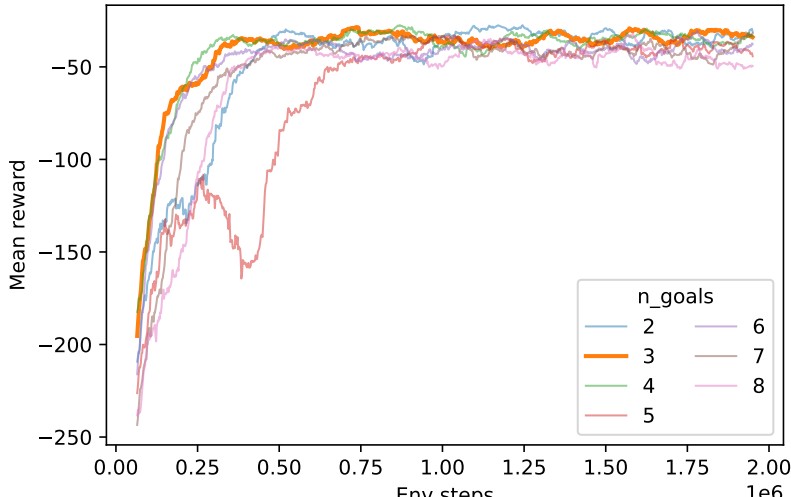

Figure 6: Episode returns for training contextual reinforcement learning. The returns are smoothed using 50 episode sliding mean.

```
FocusHasBloodClaws, FocusHasCleavers, FocusHasCripplers, FocusHasHealingGland, FocusHasVampireGland,
FocusHasFrogLegs, FocusHasPistol, FocusHasMagnum, FocusHasBlaster, FocusHasParalyzingDart,
FocusHasIronBubblegum, FocusHasHeliumBubblegum, FocusHasShell, FocusHasTrombone, UnusedExtraSense30,
UnusedExtraSense31.
```

The sub-reward functions $\boldsymbol{w}$, that act as the context, as additional features that are provided for the policy. The final dimension of the input to the policy is thus 66, the last two being reward weights for functions $r_{\text{dmg}}$ and $r_{\text{heal}}$.

### B.5.2 Action

The action space that we considered was 5-dimensional discrete space, each action being $a = \{a_1, a_2, a_3, a_4, a_5\}$. The definitions and options for each dimension are as follows:

- $a_1 \in \{-0.8, -0.4, 0.0, 0.4, 0.8\}$, the length of the step that agent takes to front or back (positive for forward)

- $a_2 \in \{-0.8, -0.4, 0.0, 0.4, 0.8\}$, the amount that the agent turns during a step (positive for turning right)

- $a_3 \in \{0.1, 0.3, 0.5, 0.7, 0.9\}$, fraction that interpolates the movement towards the focus of the agent (1.0 ignores $a_1$ and $a_2$ and just moves towards the focus)

- $a_4 \in \{0, 1, 2, 3\}$, which skill to use. 0: none, 1: melee, 2: misc(never used in our case), 3: heal.

- $a_5 \in \{0, 1, 2, 3, 4, 5, 6, 7\}$, where to focus. 0: keep current focus, 1-3: own statue and team, 4-7: enemy statue and team

### B.5.3 Reward

The reward functions $r_{\text{dmg}}$ and $r_{\text{heal}}$ were defined as following:

$$
r_{\text{dmg}} = \begin{cases} 0.1 - \min\{d_1, d_2, d_3\}, & \text{if focus enemy} \cap \text{melee} \cap d_f < 0.072 \cap |\rho_f| < 0.24 \\ -0.5 - \min\{d_1, d_2, d_3\}, & \text{if focus enemy} \cap \text{heal} \cap 0.06 < d_f < 0.072 \cap |\rho_f| < 0.2 \\ -\min\{d_1, d_2, d_3\}, & \text{otherwise,} \end{cases} \tag{10}
$$

where first condition rewards for melee attacks inflicted in sufficient proximity of an enemy and the latter penalizes for healing an enemy (without the penalty, the agent altered between healing the enemy and attacking it). The $d_f$ and $\rho_f$ stand for distance and angle to the focused agent. The minimum distance to the enemies $\min\{d_1, d_2, d_3\}$ is used for slight encouragement for movement towards the enemies to help early exploration. The reward for healing $r_{\text{heal}}$ is defined similarly, but the enemy is swapped to a teammate and there is no encouragement for reducing the distance to enemies:

$$
r_{\text{heal}} = \begin{cases} 1.0, & \text{if focus teammate} \cap \text{heal} \cap d_f < 0.072 \cap |\rho_f| < 0.24 \\ -5.0, & \text{if focus teammate} \cap \text{melee} \cap 0.06 < d_f < 0.072 \cap |\rho_f| < 0.2 \\ 0.0, & \text{otherwise.} \end{cases} \tag{11}
$$

The amount of reward were chosen so that the healing strategy and the damage reward result in similar total returns if applied consistently (one can attack more often than heal).

Since the sub-rewards relate directly to specific actions (attacking or healing) a user can take, one could here in principle attempt inferring the reward preferences also by direct inspection of the relative frequencies of these actions. That is, higher relative use of the healing action would be interpreted directly as preference for that reward. Such interpretation may, however, be severely misleading. The marginal distribution of actions is $p(a|\boldsymbol{w}) = \int p(a|s, \boldsymbol{w}) p(s|\boldsymbol{w}) ds$, where the distribution of states $p(s|\boldsymbol{w})$ depends on the policy and hence the reward. Even if preference of healing corresponds to high $p(a =' heal'|s, \boldsymbol{w})$ for many states, the preference also heavily influences the states $s$ the user visits and consequently the overall frequency of healing action is not guaranteed to be high. For instance, a healing-oriented user may try to stay further from the combat actions that strongly influences the frequency of states the attacking and healing actions are even possible.

## C  Barycentric formulation

To reason why the $m$-D navigation task is identifiable only for $m + 1$ coordinates, consider a point $p \in \mathbb{R}^m$ that resides within the set of goals $\{g_1, \ldots, g_K\}$, each goal being in the same $m$-dimensional space, $g_i \in \mathbb{R}^m$. The point $p$ is a linear combination of the goals and corresponding weights $0 < w_i < 1$ for each goal (usually called barycentric coordinates)

$$
p = w_1 g_1 + \ldots + w_K g_K, \tag{12}
$$

which lends itself for linear problem where we take into account that weights sum up to 1:

$$
\overbrace{\begin{bmatrix} 1 & \cdots & 1 \\ g_{11} & \cdots & g_{1K} \\ \vdots & & \vdots \\ g_{m1} & \cdots & g_{mK} \end{bmatrix}}^{(m+1) \times K} \begin{bmatrix} w_1 \\ \vdots \\ w_K \end{bmatrix} = \begin{bmatrix} 1 \\ p_1 \\ \vdots \\ p_m \end{bmatrix}, \tag{13}
$$

which is not solvable uniquely if $K > m + 1$.

An example of unidentifiable (non-unique) case is in Fig. 1, where a trajectory going to origin can be explained e.g. by $w = (0.5, 0.0, 0.5, 0.0)$ and $w = (0.25, 0.25, 0.25, 0.25)$.

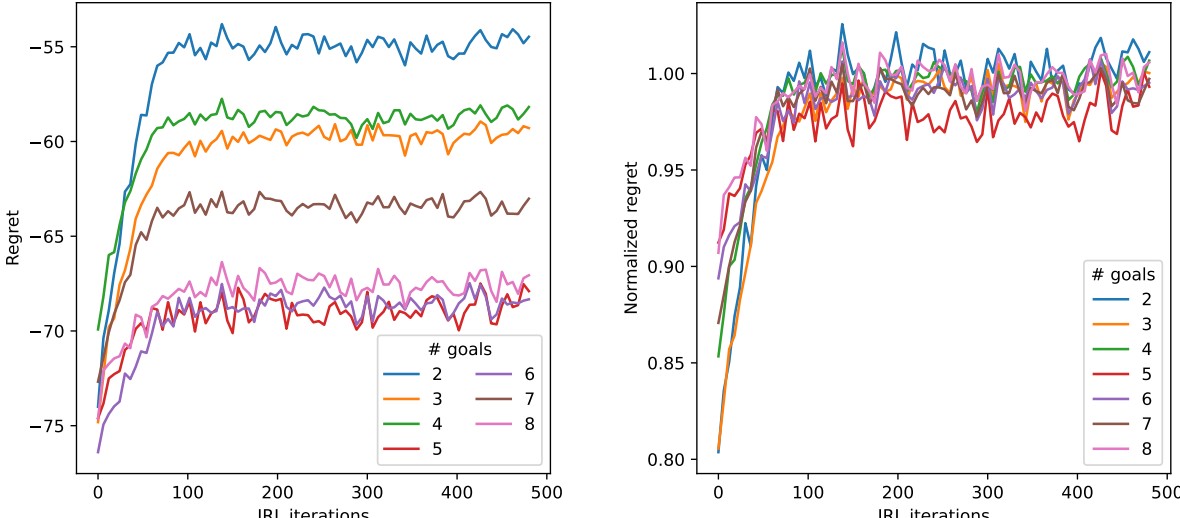

Figure 7: **Left:** Raw regret, $\mathbb{E}_{\tau \sim \pi(\cdot, \hat{\boldsymbol{w}})}[r_{\boldsymbol{w}^{(\mathrm{gen})}}(\tau)]$ as function of IRL iterations for various number of sub-rewards $K$. The curves are average regrets over the 20 expert trajectories used in Section 4.1 and over 100 trajectories generated from $\pi(\cdot, \hat{\boldsymbol{w}})$. **Right:** Normalized regret $\left(\mathbb{E}_{\tau \sim \pi(\cdot, \hat{\boldsymbol{w}})}[r_{\boldsymbol{w}^{(\mathrm{gen})}}(\tau)] / \mathbb{E}_{\tau \sim \pi(\cdot, \boldsymbol{w}^{(\mathrm{gen})})}[r_{\boldsymbol{w}^{(\mathrm{gen})}}(\tau)]\right)^{-1}$, where the performance is normalized w.r.t the generating parameters, i.e. what is achievable with cRL with the generating weights. Even though the weight recovery deteriorates for large number of goals because of non-identifiability of the solution (Figure 2), we still recover weights that lead to good policies.

## D    Regrets of Recovered Policies

In Section 4.1 we confirmed that we can recover the true $\boldsymbol{w}$ accurately. Here we provide an alternative perspective for validating the quality of the solution, by inspecting the regret curves for policies using the recovered weights. The main goal is to confirm that we can also recover good policies with the recovered weigths, but we additionally show that the IRL solution recovers useful $\hat{\boldsymbol{w}}$ also in cases where the weights are non-identifiable due to the the number of goals being too high for the environment.

For a pair of generating and recovered weights $(\boldsymbol{w}^{(\mathrm{gen})}, \hat{\boldsymbol{w}})$, we define regret as $\mathbb{E}_{\tau \sim \pi(\cdot, \hat{\boldsymbol{w}})}[r_{\boldsymbol{w}^{(\mathrm{gen})}}(\tau)]$ to measure the expected ground-truth return of a policy using the recovered weights. Figure 7 (left) plots regrets for the same $K$ considered in Figure 2, showing that for all choices the policies converge well. Figure 7 (right) confirms the policies are good, by plotting the regrets normalised so that 1 means that we are as good as a policy trained with the ground truth reward, and $< 1$ means suboptimal behavior. The normalization is done using $\left(\mathbb{E}_{\tau \sim \pi(\cdot, \hat{\boldsymbol{w}})}[r_{\boldsymbol{w}^{(\mathrm{gen})}}(\tau)] / \mathbb{E}_{\tau \sim \pi(\cdot, \boldsymbol{w}^{(\mathrm{gen})})}[r_{\boldsymbol{w}^{(\mathrm{gen})}}(\tau)]\right)^{-1}$, where we need the reciprocal because the rewards in our case are always negative.

