# OpenReview forum: "Contextual Policies Enable Efficient and Interpretable Inverse Reinforcement Learning for Populations"
_TMLR — Accepted by TMLR_

### Review · Reviewer_HHoC · 2024-01-17

**Summary Of Contributions:**

This paper considers inverse reinforcement learning (IRL) in the scenario in which there is a large population of expert demonstrations, and each individual demonstration is optimal for a reward function within a class of parameterized reward functions. The authors propose to pretrain a contextual policy on randomly sampled reward function parameters, and use the same contextual policy for gradient descent on an IRL loss for any individual. This contextual RL + IRL method (cRL+IRL) is more sample efficient than training a policy from scratch in the inner loop of IRL for each individual. The reward function is assumed to be a convex combination of component functions, where the weights are the parameters to be recovered by IRL. Experiments in an m-dimensional gridworld navigation task show that cRL+IRL recovers ground truth reward parameters on tasks up to 4 reward components, and is an order of magnitude more sample efficient than adversarial IRL. Experiments on a multi-agent team competition shows cRL+IRL can recover the reward parameters of two distinct populations of players.

**Audience:**

Yes

**Claims And Evidence:**

Yes

**Requested Changes:**

The authors should address these points to be considered for acceptance.
1. Support the claim that RL/IRL has benefits for the application setting of interest that can't be gained from simpler methods such as looking at distribution of events and features.
2. Directly state the assumption that the individual demonstrations in a population correspond to different rewards.
3. Support the claim about human behavior.
4. Justify the choice of random sampling from a degenerate Dirichlet distribution.
5. Compare to a modification of AIRL whereby the RL policy that is being trained for one individual is reused as the initialization of the policy for the next individual.

**Strengths And Weaknesses:**

Strengths:
1. This paper is relevant to some real-world IRL problems where demonstrations come from a population of individuals who may be optimizing different rewards that lie within a class of parameterized functions. In practice, when the population size is large, pretraining a contextual policy to convergence is worth the sample cost, since the policy can then be fixed and reused within IRL for each individual, without needing to train a policy from scratch each time.

Weaknesses:
1. In terms of the real-world use case of game development and understanding players' preference for different sub-rewards, it is not clear from this paper that RL and IRL are needed for this problem. One can simply look at the distribution of actions and game events.
2. In the abstract and introduction, the authors make an implicit assumption that the individual demonstrations in a population correspond to different rewards. This doesn't have to be the case in general, so omitting a clear statement of assumption invites questions.
3. The introduction states that "humans naturally do not exactly behave as reward maximizing agents". This is a strong claim without justification or reference.
4. When training the contextual policy, the method randomly selects context $\omega$ from a degenerate Dirichlet distribution. This choice is insufficiently motivated: is this the best thing to do if the population of demonstrations follows another distribution? How would the proposed method accommodate this scenario?
5. The use of RL and IRL for the application settings of interest in this paper isn't fully justified. The paper motivates the setting of IRL on a population by stating the need for game developers to understand the game players' preferences for optimizing different sub-rewards. Without using IRL, one can still understand their preferences based on the distribution of game features and actions taken by players, e.g. in the Derk experiment, look at the distribution of damage and healing actions, which shows players' preference for the two reward components. That is easy to do, so it's not clear why and when one would need RL/IRL and the proposed method.
6. Besides comparing to vanilla adversarial IRL (AIRL), the authors should compare to a modification of AIRL whereby the RL policy that is being trained for one individual is reused as the initialization of the policy for the next individual.

---

> ### Author Response · Authors · 2024-02-16
>
> Thank you for the review! We would like to answer to the requested changes below and with the revised manuscript:
>
> ***Support the claim that RL/IRL has benefits for the application setting of interest that can't be gained from simpler methods such as looking at distribution of events and features.***
>
> The simpler methods used in the field are (on a general level) based on inspecting summary statistics, like the overall distribution of the actions $p(a)$. The motivation for using IRL formulation is that it allows moving beyond this limited perspective. First, the RL formulation explicitly builds on the *conditional* distribution $p(a|s)$ of the actions for different states, where the state $s$ is often (and in our examples) high-dimensional and/or continuous. Second, RL considers long-term rewards over *sequences of actions*. We cannot imagine any obvious ways of capturing these characteristics by direct inspection of the distributions, whereas IRL provides a natural approach that is already used in the field as confirmed by several citations in our work.
>
> We now explicate these characteristic properties in Introduction to make the limitations of direct analysis clear.
>
>
> ***Directly state the assumption that the individual demonstrations in a population correspond to different rewards.***
>
> We now explicitly mention in Section 2.3 that we assume each individual *user* to behave according to a single reward function. Note that we can have multiple demonstrations from a single user.
>
> ***Support the claim about human behavior.***
>
> Indeed the claim was unnecessary and we re-wrote it to better capture the intended meaning: whether or not humans behave as reward maximizing agents, modeling them as such can be useful. We also added a reference to a paper discussing the issue more broadly.
>
> ***Justify the choice of random sampling from a degenerate Dirichlet distribution.***
>
> We did early experimentation of training the contextual policy with weights sampled from a non-degenerate Dirichlet distribution as well, but found that it easily gets stuck to local optimum. See our answer to the second question of Reviewer 1a9e.
>
> ***(From weaknesses:) This choice [sampling from degenerate Dirichlet] is insufficiently motivated: is this the best thing to do if the population of demonstrations follows another distribution? How would the proposed method accommodate this scenario?***
>
> Even though we train the cRL with weights sampled from a degenerate distribution, we always evaluate the solution on demonstrations that follow a different distribution. That is, we empirically confirm it works in such scenarios. We now mention this already in Section 3.2.3.
>
> For experiment 1, we sampled the true rewards $\boldsymbol{w}$ from a uniform distribution (see Section 4.1.2) and for experiment 2 from a manually defined informative distribution (see Section 4.2.3).
>
> ***Compare to a modification of AIRL whereby the RL policy that is being trained for one individual is reused as the initialization of the policy for the next individual.***
>
> This is an interesting and clever idea. We are not aware of specific methods proposed for this, but now added a simple baseline that implements the basic idea. It does not specifically initialize the model with the \textit{previous user}, but to a sensible checkpoint from earlier experiment, and nevertheless quantifies directly whether there is value from starting from a solution pre-trained with other users.
>
> Figure 2 (right) presents the new results, showing that the initial performance of AIRL indeed increases, but the overall convergence does not improve much and the variant remains considerably slower than the proposed method.

---

> ### Comment · Reviewer_HHoC · 2024-03-05
> **Acknowledgement of reply**
>
> I appreciate the author's reply, amendments to the text, and the additional experiment with AIRL-warmstart.
>
> > First, the RL formulation explicitly builds on the conditional distribution
>  of the actions for different states, where the state
>  is often (and in our examples) high-dimensional and/or continuous. Second, RL considers long-term rewards over sequences of actions. We cannot imagine any obvious ways of capturing these characteristics by direct inspection of the distributions
>
> Even so, the main focus at the end is not on the policy but rather on the reward function weights, e.g. weights for damage or healing in the Derk environment. When I look at the weights to understand a user's behavior, I don't need to think about any conditional distribution. It seems I can understand the user's behavior just as well by merely looking at the relative occurrence frequency of damage actions versus healing actions, without going through IRL. So my original question was asking about the difference in practice.
>
> Edit: the authors may consider saying this method is necessary to get the weights of a more complex reward function that has state-dependent conditional terms, and this info cannot be acquired from looking at action distributions alone.

---

> > ### Author Response · Authors · 2024-03-06
> >
> > We would like to still clarify this from two perspectives:
> > 1. Even though in the Derk example the rewards correspond to specific actions, this would not be the case in general. We now added a remark clarifying this in Introduction (2nd to last paragraph). As a concrete example, in the m-D navigation environment of Section 4.1 the reward weights correspond to the different *goals* whereas the actions are *movement directions and lengths*. Looking at the distribution of the actions is simply not sensible, and looking at the relative frequencies of reaching a particular goal would not work either as the users typically do not reach any of them.
> > 2. In the Derk environment a healing-oriented character is actually best identified by their *movement patterns* (they attempt to stay close to other characters that are involved in combat), not explicitly by increased use of the healing action. They will still use the attack action when they end up in direct combat to prolong the game and increase chances of future rewards, and there are no guarantees that the relative ratio of healing vs attack would be higher (even though it typically is) than for the attack-focused users. We now clarify in the end of Appendix B.5.3 that the marginal distribution of actions $p(a) = \int p(a|s) p(s) ds$ heavily depends on the states the user visits during the trajectory, and that direct inspection of the marginal distribution can be misleading.

---

> > > ### Comment · Reviewer_HHoC · 2024-03-06
> > > **Acknowledged**
> > >
> > > These remarks are helpful, thank you.

---

### Review · Reviewer_1a9e · 2024-01-19

**Summary Of Contributions:**

This paper proposes a highly scalable approach for performing inverse RL on a population of users. The idea is to use contextual RL to pre-train an optimal policy that works for any reward function and then use this to improve the scalability of IRL by removing the need to run RL iterations during reward inference. The approach is shown to be much more efficient than AIRL and to work well on both a navigation and battle video game environment.

**Audience:**

Yes

**Claims And Evidence:**

Yes

**Requested Changes:**

would having state and action spae change on context break everything?

why not try selecting weighted combinations of multiple sub-rewards during training? It would be nice to see an example of this and how it compares with the current approach. It isn't clear to me that interpolation will always work if not explicitly trained on some examples of interpolation.

When analyzing the complexity in 3.2.4, what is I?

In 4.1.2, It's not clear to me that the action space is really high-dimensional. Isn't it 2-d since actions are directions to move? I don't understand how you get 80 or what m is in this domain. what is the nuisance part of the action and why is it included?

Critical: I'm confused about alpha in the Dirichlet model. Early in the paper it says alpha ->0, but then in 4.1.2 alpha = 1.0. Why the discrepancy?

4.1.3 Typo "pre-taining" -> "pre-training"

Critical: In Figure 2 it is unclear what the difference is between Recovered and Mean pred in the center plot.

Critical: An RL performance curve on the true reward weights for the experiment results shown in Fig 2 is missing. For example, even though some demonstrations may have multiple weight vectors that describe then (especially for large K), it shouldn't matter in terms of policy performance. I would like to see some measure of regret wrt the user's true reward and the behavior obtained via the IRL+cRL approach proposed in this paper. I think a regret plot would show the full picture since sometimes you can get close to the true reward but still learn a bad policy and sometimes you don't have to learn the exact same reward to get the same policy.

**Strengths And Weaknesses:**

Strengths:
+ Good motivation. I like the application to play-testing.
+ Nice idea to use cRL to speed up IRL. It's simple and seems to work well.
+ Good comparison with AIRL.

Weaknesses:
- Some parts of the paper were unclear and hard to follow.
- Only using 2 subrewards for the complex benchmark seems limiting. It would have been nice to see a more expressive reward.
- Only 2 environments.
- No discussion of actual regret in terms of policy performance, just a comparison of reward similarity.

---

> ### Author Response · Authors · 2024-02-16
>
> Thank you for the review! We would like to answer to the requested changes below and with the revised manuscript:
>
> ***would having state and action space change on context break everything?***
>
> We remind that the context in our case refers *only* to the reward preferences $\boldsymbol{w}$ of the user and hence cannot influence the state and action spaces that are parts of the game itself.
>
> From a technical perspective, cRL could be trained also in environments that depend on the context but it is out of the scope here.
>
> ***why not try selecting weighted combinations of multiple sub-rewards during training? It would be nice to see an example of this and how it compares with the current approach. It isn't clear to me that interpolation will always work if not explicitly trained on some examples of interpolation.***
>
> We did early experimentation of training a contextual policy with sampling weighted combinations (instead of one-hot vectors). It works, but is more prone to getting stuck to local optimums during cRL training and is thus clearly suboptimal for some goals. What happened was that every sampled weight gave some reward for every goal (sometimes more, sometimes less, depending on the weight sampled) and the policy  chose one or few goals to go to and then stuck with those without ever considering other goals.
> Explicitly using $\boldsymbol{\alpha}=\boldsymbol{0}$ also removes one hyperparameter, which is always beneficial in RL contexts. Note that we empirically validate that the cRL model does work for general $\boldsymbol{w}$.
>
> We added a footnote outlining the above observation to empirically motivate our choice, but decided not to add an explicit comparison since we do not consider this to be a central part of the method. It is possible that in some environments an efficient interpolation requires to explicitly train the policy on some reward parameters from the "in-between" zone to improve the interpolation. This is an interesting research area and best studied in works that focus on the cRL training itself.
>
> ***When analyzing the complexity in 3.2.4, what is I?***
>
> $I$ is the number of IRL iterations that is referred to in algorithms 1 and 2. We now mention it also in Section 3.2.4.
>
>
> ***In 4.1.2, It's not clear to me that the action space is really high-dimensional. Isn't it 2-d since actions are directions to move? I don't understand how you get 80 or what m is in this domain. what is the nuisance part of the action and why is it included?***
>
> We specifically crafted a high-dimensional discrete action space where the agent chooses both direction and stride length (can be thought of as speed) from a Cartesian product of alternatives for both, as explained in Section 4.1.2.  Even though the alternative actions have intuitive relationships for a human, the algorithm is not given any information about their meanings and only sees a high-dimensional discrete action space and needs to cope with that.
>
>
> The $m$ denotes the dimensionality of the environment. The illustrations in Figure 1 were for $m=2$ so that the environment can be drawn visually, but we ran the real experiments with $m=5$. Already for $m=2$ we would have 16 possible actions because of four alternative directions ([up, down, left, right]) and four possible stride lengths. For $m=5$ we have $10$ possible directions and again $4$ stride alternatives, corresponding to $40$ choices.
>
> Finally, we added a second set of $40$ choices that are actually identical in terms of behavior by taking a further Cartesian product with a dummy nuisance variable, to mimic cases where the game may offer multiple actions that lead to the same state (this was originally motivated by the Derk game having some actions that were disabled in the environment). This doubles the final action space to $80$ options.
>
> (Response continues in the next message)

---

> > ### Author Response · Authors · 2024-02-16
> >
> > ***Critical: I'm confused about alpha in the Dirichlet model. Early in the paper it says alpha $\rightarrow$ 0, but then in 4.1.2 alpha = 1.0. Why the discrepancy?***
> >
> > Section 3.2.3 explains that we use $\boldsymbol{\alpha} \rightarrow 0$ to sample $\boldsymbol{w}$ for *training the contextual policy*. In Section 4.1.2 we used $\boldsymbol{\alpha}_{\text{gen}}  = \boldsymbol{1.0}$ for generating *the expert behaviors that are used as data in the experiment*. For each sampled weight $\boldsymbol{w}$ we generate expert demonstrations using the corresponding contextual policy.
> >
> >
> > To avoid confusion, we now do not introduce a separate symbol for the parameter used while generating the expert trajectories, but simply write that the generating weights are sampled from $\text{Dir}(\bf{1.0})$ in Section 4.1.2.
> >
> > Even though the choice of $\boldsymbol{\alpha} \rightarrow 0$ for training the cRL was done primarily to improve the cRL learning, the discrepancy between the training cases and the ground truth weights used for generating the data additionally serves as a test for a covariate shift. That is, we explicitly show that the cRL solution works on weights not seen during the training.
> >
> >
> > ***Critical: In Figure 2 it is unclear what the difference is between Recovered and Mean pred in the center plot.***
> >
> > The labels were misleading. We now re-labeled *Mean pred* to be *Baseline*, to better reflect the purpose of that curve. Here *Recovered* refers to the result of our method whereas *Baseline* tells the accuracy of a maximally naive estimator that sets all weights at $1/K$, and is shown only to provide a natural scale for the results.
> >
> > ***Critical: An RL performance curve on the true reward weights for the experiment results shown in Fig 2 is missing. For example, even though some demonstrations may have multiple weight vectors that describe then (especially for large K), it shouldn't matter in terms of policy performance. I would like to see some measure of regret wrt the user's true reward and the behavior obtained via the IRL+cRL approach proposed in this paper. I think a regret plot would show the full picture since sometimes you can get close to the true reward but still learn a bad policy and sometimes you don't have to learn the exact same reward to get the same policy.***
> >
> > This is a good suggestion as an alternative perspective for interpreting the solution. Even though our main goal is in estimating the user weights, it certainly makes sense to check also that the recovered policies are good.
> >
> > We now included two regret plots in Appendix D, showing the expected returns *for the true user weights* when running a *policy using the recovered weights*, again for a range of $K$. We show both the raw regrets (that depend on $K$) as well as regrets normalised by the returns of *a policy using the true user weights*.
> > Please refer to the Appendix D for exact formulation.
> >
> >
> > The main result is, like you hypothesized, that the policies using the recovered weights reach the same expected return that we would get with the ground truth weights, also for the cases of high $K$ where we cannot directly identify the true rewards (weights).

---

> ### Author Response · Authors · 2024-03-21
>
> Dear Reviewer 1a9e,
>
> Thank you for taking the time to review our paper. We hope that you find our response helpful for addressing the concerns you pointed out. If not, would you be kind enough to provide additional comments on our response, so that we can clarify it and improve the manuscript.
>
> Best regards,
>
> The Authors

---

### Review · Reviewer_mwio · 2024-02-10

**Summary Of Contributions:**

The paper proposed to combine Inverse Reinforcement Learning (IRL) with Contextual Reinforcement Learning (cRL). Starting from the idea that a large variety of IRL approaches is required to solve the RL problem for every candidate reward function, the authors propose to address this task with cRL, where the context represented by the reward weights recovered from IRL, determines the reward function. In such a way, the experience collected for learning the optimal policy with one reward function can be reused in the future. The author focuses on linearly parametrized reward function for promoting interpretability of the output of the IRL procedure. Experimental validation in an m-D navigation task and on the Derk game is provided.

**Audience:**

Yes

**Broader Impact Concerns:**

None.

**Claims And Evidence:**

No

**Requested Changes:**

Please refer to "Strengths And Weaknesses". Anyway my main concern is the first one about the appropriateness of the methodological approach.

**Strengths And Weaknesses:**

**Strengths**
- The idea of using cRL to exploit the similarities between reward functions and re-use implicitly the experience collected in the past to make the full IRL pipeline more sample efficient is of interest.
- The experimental results are promising. In the first experiment, the sample efficiency compared with AIRL is visible. Moreover, the authors succeed in showing that the proposed approach is able to recover the original reward weights.

**Weaknesses**
- The first methodological concern is about the appropriateness of using cRL when the final task is IRL. While I accept the informal rationale that cRL is able to save samples thanks to the pre-trained policy, I am not sure if similar rewards (more formally, reward functions characterized by weight vectors that are close, representing the context) lead to similar optimal policies. Surely, they will lead to similar value functions, but, in IRL our goal is to discriminate whether a reward function allows learning the expert's policy (we do not really care about the value functions). Can the authors specify whether they have a more formal justification for the appropriateness of using cRL for IRL?

- The experimental validation is quite limited for what concerns the comparison with state-of-the-art IRL approaches. If the ultimate goal of the paper is to make IRL more sample efficient, I believe that a comparison with IRL approaches that do not require solving the forward RL problem should be performed. For instance:

Ramponi, G., Likmeta, A., Metelli, A. M., Tirinzoni, A., & Restelli, M. (2020, June). Truly batch model-free inverse reinforcement learning about multiple intentions. In International conference on artificial intelligence and statistics (pp. 2359-2369). PMLR.

Klein, E., Piot, B., Geist, M., & Pietquin, O. (2013). A cascaded supervised learning approach to inverse reinforcement learning. In Machine Learning and Knowledge Discovery in Databases: European Conference, ECML PKDD 2013, Prague, Czech Republic, September 23-27, 2013, Proceedings, Part I 13 (pp. 1-16). Springer Berlin Heidelberg.

- It is not completely clear whether what is proposed in the paper is an *abstract general pipeline* or a *specific combination* of an IRL algorithm with a cRL algorithm. The authors, indeed, in Section 2.3 propose a specific maximum-likelihood formulation of the IRL problem (which is a choice of an IRL approach), and in Section 3.2.3 a specific algorithm for cRL (Woillemont et al. 2021). The authors should clarify in the presentation whether these choices are compulsory or can be replaced with other IRL and cRL algorithms (and with which modifications).

---

> ### Author Response · Authors · 2024-02-16
>
> Thank you for the review! We would like to answer to the requested changes below and with the revised manuscript:
>
> ***The first methodological concern is about the appropriateness of using cRL when the final task is IRL. While I accept the informal rationale that cRL is able to save samples thanks to the pre-trained policy, I am not sure if similar rewards (more formally, reward functions characterized by weight vectors that are close, representing the context) lead to similar optimal policies. Surely, they will lead to similar value functions, but, in IRL our goal is to discriminate whether a reward function allows learning the expert's policy (we do not really care about the value functions). Can the authors specify whether they have a more formal justification for the appropriateness of using cRL for IRL?***
>
> We would like to clarify that our main goal differs slightly from the IRL goal you stated. In many other works the reward is estimated as an intermediate step to learn policies with the estimated rewards, whereas our main goal is in estimating the reward itself, in order to interpret the intent behind the behavior (e.g. a player is healing-oriented). Being able to learn a good policy (e.g. for simulating how a particular player would play in a new environment) is interesting also for us but it is not necessary for our main claims.
>
> Our main results empirically demonstrate that we can recover the generating rewards also when using cRL, and in the revised version we additionally show (in response to Reviewer 1a9e) that the regrets for policies using the recovered weights align with the regrets of a policy using the true user weights, even when we cannot idenfity the weights. We believe this is solid empirical evidence for the claims we make.
>
> In general, a small change in the reward can change the policy notably (for instance, switching from slight preference for reward A over B to slight preference for B changes the optimal policy if the sub-rewards are mutually exclusive). However, this would be the case for all IRL algorithms and is not related to our use of cRL.
>
> ***The experimental validation is quite limited for what concerns the comparison with state-of-the-art IRL approaches. If the ultimate goal of the paper is to make IRL more sample efficient, I believe that a comparison with IRL approaches that do not require solving the forward RL problem should be performed. For instance: Ramponi et al. (2020), Klein et al. (2013).***
>
>
> We thank for the pointer for Ramponi et al. (2020). It is indeed closely related but we had missed it. We now discuss it in the Introduction when explaining other IRL approaches. However, it solves a different problem and we cannot empirically compare against it.
> Our main goal is to discover *individual* user-specific rewards, whereas Ramponi et al. (2020) considers a scenario where the population is explicitly assumed to consist of *clusters of users, with all users in a cluster sharing the exact same reward*. Even though we use a clustering setup in the Derk experiment, the clustering was done solely for the purpose of simplifying the data generation and visualizations -- our method still infers separate rewards for every user. We now clarify this in Section 4.2.1.
>
> Also note that Ramponi et al. is an offline method where the agents do not interact with the environment during training, which is also a major technical difference that would make explicit comparisons meaningless. For instance, we measure the convergence speed in terms of the number of environment interactions.
>
> Regarding empirical comparisons in general, we now added a new baseline (see response to Reviewer HHoC for details) that attempts to speed up AIRL in an alternative way of sharing information across individual users. The new baseline initializes user-specific models better based on information from other users and is slightly better, but still considerably slower than the proposed method and does not change the conclusions.
>
>
> ***It is not completely clear whether what is proposed in the paper is an abstract general pipeline or a specific combination of an IRL algorithm with a cRL algorithm. The authors, indeed, in Section 2.3 propose a specific maximum-likelihood formulation of the IRL problem (which is a choice of an IRL approach), and in Section 3.2.3 a specific algorithm for cRL (Woillemont et al. 2021). The authors should clarify in the presentation whether these choices are compulsory or can be replaced with other IRL and cRL algorithms (and with which modifications).***
>
> The paper proposes *abstract general pipeline* for cRL + IRL, but we naturally needed to create a specific instance to conduct the empirical experiments. We now clarified this in Sections 3.2.1 and 3.2.3.

---

> ### Author Response · Authors · 2024-03-21
>
> Dear Reviewer mwio,
>
> Thank you for taking the time to review our paper. We hope that you find our response helpful for addressing the concerns you pointed out. If not, would you be kind enough to provide additional comments on our response, so that we can clarify it and improve the manuscript.
>
> Best regards,
>
> The Authors

---

### Decision · Action_Editor_WUJb · 2024-05-20

**Recommendation:** Accept with minor revision

**Comment:**

The paper presents a novel approach that combines Inverse Reinforcement Learning (IRL) with Contextual Reinforcement Learning (cRL) to enhance the efficiency and scalability of IRL. The authors propose pre-training a contextual policy that can adapt to various reward functions, thereby reusing past experience and improving sample efficiency. The method is tested on an m-dimensional navigation task and the Derk game, showing promising results in recovering reward weights and outperforming traditional adversarial IRL (AIRL) in terms of sample efficiency.

The innovative use of cRL in IRL is praised for its practical applicability, especially in real-world scenarios such as game development and user behavior analysis. The experiments demonstrate significant improvements in sample efficiency, supporting the method's effectiveness and versatility.

The reviewers have raised some concerns about the methodological justification of using cRL for IRL. The assumption that similar reward functions lead to similar optimal policies needs more formal support. The experimental comparisons with state-of-the-art IRL methods are considered insufficient, and the scope of the experiments is limited to two environments with not sufficiently expressive rewards. Some parts of the paper are unclear or poorly explained, and the assumptions about demonstrations and human behavior lack explicit statements and support.

The authors have properly addressed most of these issues in their rebuttals. Still, I have pointed out further issues about handling the constraint in the optimization problem and the identifiability problem.
The authors modified their paper to explain how they project the solution after each gradient update and briefly mention the limitations of their approach regarding the identifiability problem.
I would like the authors to discuss this problem more: according to the basis functions used for the reward, their optimization problem can have multiple optimal solutions. Searching for the weights that minimize the difference between the expert performance and the one of the optimal contextual policy can lead to optimal solutions (difference equal to zero) where the difference in the single reward components can be significant. The authors should highlight these problems and suggest how they could be addressed.

Furthermore, the use of contextual policies in which the context depends only on the reward function should actually be referred to as Multi-Objective Reinforcement Learning and the corresponding literature should be considered.
An example of papers that learn policy parameterized with respect to the reward function are:

Castelletti, A., Pianosi, F., & Restelli, M. (2012, June). Tree-based fitted Q-iteration for multi-objective Markov decision problems. In The 2012 International Joint Conference on Neural Networks (IJCNN) (pp. 1-8). IEEE.

Parisi, S., Pirotta, M., & Restelli, M. (2016). Multi-objective reinforcement learning through continuous Pareto manifold approximation. Journal of Artificial Intelligence Research, 57, 187-227.

Yang, R., Sun, X., & Narasimhan, K. (2019). A generalized algorithm for multi-objective reinforcement learning and policy adaptation. Advances in neural information processing systems, 32.

**Audience:**

The paper is of interest to a part of TMLR's audience.

**Claims And Evidence:**

The claims of the paper are supported by empirical evidence, but the authors should highlight the limitations better.

---

> ### Author Response · Authors · 2024-06-05
>
> We thank the AE for the decision. We have deanonymized the manuscript and included the suggested discussion. More specifically:
>
>  - Introduction's 7th paragraph now includes discussion of Multi-Objective RL along with references. We thank for pointing us to this important and relevant literature.
>  - Section 3.2.4 discloses and discusses non-identifiability of the softmax reparametrization.
>  - Section 5's 3rd paragraph points to Appendix A, that details the degenerate solution of the loss function in mathematical terms.
>
> We are grateful for reviewers who improved the manuscript by providing invaluable comments.